# FLOW STRAIGHT AND FAST IN HILBERT SPACE: FUNCTIONAL RECTIFIED FLOW

**Jianxin Zhang**
Electrical Engineering and Computer Science
University of Michigan
Ann Arbor, MI 48109
`jianxinz@umich.edu`

**Clayton Scott**
Electrical Engineering and Computer Science
University of Michigan
Ann Arbor, MI 48109
`clayscot@umich.edu`

## ABSTRACT

Many generative models originally developed in finite-dimensional Euclidean space have functional generalizations in infinite-dimensional settings. However, the extension of rectified flow to infinite-dimensional spaces remains unexplored. In this work, we establish a rigorous functional formulation of rectified flow in an infinite-dimensional Hilbert space. Our approach builds upon the superposition principle for continuity equations in an infinite-dimensional space. We further show that this framework extends naturally to functional flow matching and functional probability flow ODEs, interpreting them as nonlinear generalizations of rectified flow. Notably, our extension to functional flow matching removes the restrictive measure-theoretic assumptions in the existing theory of Kerrigan et al. (2024). Furthermore, we demonstrate experimentally that our method achieves superior performance compared to existing functional generative models.

## 1 INTRODUCTION

Generative modeling has witnessed significant advancements, with methods such as flow matching (Lipman et al., 2023), diffusion models (Song and Ermon, 2020; Ho et al., 2020; Song et al., 2021), and rectified flows (Liu et al., 2022) achieving state-of-the-art performance across various data types, including audio Kong et al. (2021); Goel et al. (2022), image Dhariwal and Nichol (2021); Kang et al. (2023), and video Ho et al. (2022); Saharia et al. (2022).

Many of these techniques, originally developed in finite-dimensional Euclidean spaces, have been extended to infinite-dimensional settings, as seen in functional flow matching (Kerrigan et al., 2024), functional GAN (Rahman et al., 2022) and functional diffusion models (Franzese et al., 2023; Sun et al., 2023; Bond-Taylor and Willcocks, 2024; Na et al., 2025). Functional models enable variable-resolution generation and simple, modality-agnostic architectures, making them effective across images, 3D objects, and scientific data (Dupont et al., 2022; Kerrigan et al., 2024). They can also improve training speed, sampling efficiency, and memory use (Bond-Taylor and Willcocks, 2024; Franzese et al., 2023). However, despite the success of rectified flows in finite dimensions, their functional generalization remains an open problem.

In this work, we develop a mathematically rigorous framework for rectified flows in general separable Hilbert spaces. We further show that this framework naturally extends to proposed models of functional flow matching Kerrigan et al. (2024) and function probability flow ODEs Na et al. (2025) in Hilbert spaces, interpreting them as nonlinear generalizations of rectified flow. Notably, our approach removes the restrictive measure-theoretic assumptions required in Kerrigan et al. (2024). This theoretical foundation offers a unified lens for functional generative models. Additionally, we empirically validate functional rectified flow models and demonstrate their superior performance compared to other functional generative models.

Our contributions can be listed as follows:

- We develop a theoretical framework that extends rectified flow to a general separable Hilbert spaces.

- We provide the generative insight that several existing ODE-based functional generative models, including Functional Flow Matching (FFM) (Kerrigan et al., 2024), functional PF-ODEs (Na et al., 2025), and our Functional Rectified Flow, naturally arise as instances of a single unified formulation.

- We provide empirical validation across images and PDE dynamics, demonstrating that FRF achieves competitive or superior performance compared with prior functional models under matched architectures.

The remainder of the paper is organized as follows. Section 2 reviews related work on rectified flows and functional generative models. In Section 3, we present our main theoretical result, which lifts rectified flows to general Hilbert spaces. Section 4 discusses connections to functional flow matching and functional probability flow ODE. Architectural choices for functional rectified flow models are discussed in Section 5. Experimental results across several domains are reported in Section 6. Finally, Section 7 concludes with a summary of findings and a discussion of limitations and future directions.

## 2  RELATED WORK

Generative models have significantly advanced in recent years, with methods such as Generative Adversarial Networks (GANs), diffusion models, and flow matching achieving state-of-the-art results. GANs, introduced in Goodfellow et al. (2014), leverage adversarial training to generate high-quality samples from complex data distributions. Diffusion models, based on stochastic differential equations, iteratively transform from noise to data via a score function approximated by minimizing a quadratic objective (Vincent, 2011), demonstrating strong generative capabilities (Ho et al., 2020; Song and Ermon, 2020; Song et al., 2021). Flow matching constructs a path of conditional Gaussian distributions to interpolate the data distribution and a reference distribution (Lipman et al., 2023; Chen et al., 2018). These approaches have achieved state-of-the-art performance across diverse modalities, including audio (Kong et al., 2021; Goel et al., 2022), images (Dhariwal and Nichol, 2021; Kang et al., 2023), and video (Ho et al., 2022; Saharia et al., 2022). Recent advances (Albergo and Vanden-Eijnden, 2023; Rout et al., 2024) further show that deterministic flows and SDE-based diffusion models can be understood within a unified framework.

Rectified flows, introduced by Liu et al. (2022), offer a deterministic alternative to stochastic generative models by constructing straight transport paths between source and target distributions. In contrast to diffusion models, which rely on stochastic sampling, rectified flows enable more efficient and interpretable generation with reduced computational overhead. The associated straightening effect has been shown to facilitate high-quality generation with very few sampling steps (Lee et al., 2024). Further theoretical development, such as its connection to Optimal Transport, has been explored in Liu (2022). Recent advances have extended rectified flow methods to a broad range of generative tasks, including image generation and editing (Zhu et al., 2024; Rout et al., 2024; Ma et al., 2024; Yang et al., 2024; Dalva et al., 2024), 3D content creation (Go et al., 2024), text-to-speech synthesis and editing (Guan et al., 2024; Liu et al., 2024b; Guo et al., 2024; Yin et al., 2025), audio reconstruction (Yuan et al., 2025; Liu et al., 2024c), video generation (Wu et al., 2024), and multi-modal generative modeling (Li et al., 2024; Liu et al., 2024a). Despite this growing popularity, existing rectified flow models are still constrained to finite-dimensional spaces. In this work, we address this limitation by extending rectified flow to general Hilbert spaces, thereby enabling modeling in infinite-dimensional function spaces.

A key motivation for studying functional generative models is that many data sources are inherently functional—such as snapshots of time series or solutions to partial differential equations. A prominent example is neural stochastic differential equations (Neural SDEs), where neural networks are trained to model path-valued random variables that solve SDEs (Kidger et al., 2021). However, these models typically assume that the data follows an underlying SDE structure, as in financial time series (Zhang et al., 2025; Issa et al., 2023), which limits their general applicability. On the other hand, representing finite-dimensional data as continuous functions offers several advantages: it naturally supports variable resolution, accommodates diverse data modalities using simple architectures, and enhances memory efficiency (Dupont et al., 2022). Motivated by these benefits, recent works have extended generative modeling beyond finite-dimensional Euclidean spaces to functional settings. Dutordoir et al. (2023) and Zhuang et al. (2023) adapt existing diffusion models to functional data by conditioning on discretized pointwise evaluations.

A more general direction involves defining stochastic processes directly in infinite-dimensional Hilbert spaces, leading to the development of functional diffusion models (Kerrigan et al., 2023; Franzese et al., 2023; Bond-Taylor and Willcocks, 2024; Mittal et al., 2022; Lim et al., 2023a;b; Pidstrigach et al., 2024; Hagemann et al., 2023; Na et al., 2025). These works extend score-based methods by studying diffusion processes over function spaces. Similarly, functional flow matching (Kerrigan et al., 2024) generalizes the flow matching framework of Lipman et al. (2023) to infinite-dimensional settings. However, as noted in their work, the analysis of Kerrigan et al. (2024) relies on strong measure-theoretic assumptions that are often difficult to verify in practice. In contrast, we extend rectified flow to Hilbert spaces under more tractable and verifiable conditions, providing a rigorous and broadly applicable foundation for functional generative modeling.

## 3 RECTIFIED FLOWS ON HILBERT SPACE

In this section, we extend rectified flows to infinite-dimensional Hilbert spaces, demonstrating the fundamental property of marginal distribution preservation. Let $\mathcal{H}$ be a separable Hilbert space. Given a probability triplet $(\Omega, \mathcal{S}, \mathbb{P})$, denoting the sample space, sigma algebra, and probability measure, we take the perspective that the law of a stochastic process $\mathbb{X} : [0,1] \times \Omega \to \mathcal{H}$ is the distribution of the function-valued random variable defined as $\omega \mapsto \mathbb{X}(\cdot, \omega)$. For simplicity, we denote the value of $\mathbb{X}$ at time $t \in [0,1]$ by $X_t$ and the associated marginal measure by $\mu_t$. A random process $\mathbb{X}$ will thus also be denoted $\{X_t\}_{t \in [0,1]}$ or simply $\{X_t\}_t$. Unless otherwise specified, $\langle \cdot, \cdot \rangle$ refers to the inner product in $\mathcal{H}$, and the norm $\|\cdot\|$ refers to its induced norm.

### 3.1 DEFINITIONS OF RECTIFIABLE AND RECTIFIED FLOWS

Given a process $\mathbb{X}$, the induced rectified flow is a random process $\{Z_t\}_{t \in [0,1]}$ that is defined in terms a quantity called the expected velocity of $\mathbb{X}$.

**Definition 1.** *Let $\mathbb{X} = \{X_t\}_{t \in [0,1]}$ be a pathwise continuously differentiable random process on a separable Hilbert space $\mathcal{H}$. The* expected velocity *of $\mathbb{X}$ is defined as*

$$v^{\mathbb{X}}(t, x) = \mathbb{E}\Big[\dot{X}_t \mid X_t = x\Big], \quad \forall x \in \text{supp}(X_t), \forall t \in [0,1]$$

*and is set to zero if $x \notin \text{supp}(X_t)$. Here, $\dot{X}_t$ denotes the time derivative of $X_t$, and $\text{supp}(X_t)$ the support of the random variable $X_t$.*

Let $C^1$ denote the space of continuously differentiable functions. Pathwise continuous differentiability is not a data assumption but a benign restriction on the design of $\mathbb{X}$, i.e., $t \mapsto X_t$ needs to be $C^1$ with probability 1. For an interpolation process $X_t = \alpha_t X_1 + \beta_t X_0$, the pathwise continuous differentiability condition is satisfied by taking both $\alpha_t$ and $\beta_t$ to be real-valued $C^1$ functions.

An initial value problem (IVP) is a differential equation with a specified initial condition for the unknown function. To ensure the well-posedness of rectified flows in Hilbert space, we assume that the corresponding IVP admits a unique solution and a continuous solution map.

**Assumption 2.** *Given a function $v : [0,1] \times \mathcal{H} \to \mathcal{H}$, the initial value problem*

$$z(t) = u + \int_0^t v(s, z(s))ds, \quad u \in \mathcal{H}, \tag{1}$$

*admits a unique solution $z(\cdot) \in C^1([0,1]; \mathcal{H})$. Furthermore, the solution map $\Phi : \mathcal{H} \to C^1([0,1]; \mathcal{H})$ mapping from the initial value to the solution path, defined by $\Phi(u) = z$, is a continuous mapping from $(\mathcal{H}, \|\cdot\|_{\mathcal{H}})$ to $(C^1([0,1]; \mathcal{H}), \|\cdot\|_{\infty})$.*

In Proposition 18 in Appendix, we show that Assumption 2 holds under some standard Lipschitz condition. We emphasize that Proposition 18 is presented as a convenient sufficient condition rather than as a core assumption, since Assumption 2 can in principle hold under other regularity conditions. In the following assumption, we enforce that $v^{\mathbb{X}}(t, x)$ has a finite integral in time to ensure that the process does not exhibit pathological behavior, such as infinite total drift.

**Assumption 3.** *Given a pathwise continuously differentiable random process $\mathbb{X} = \{X_t\}_{t \in [0,1]}$,*

$$\int_0^1 \int_{\mathcal{H}} \|v^{\mathbb{X}}(t, x)\| d\mu_t(x) dt < \infty, \text{ and } \mathbb{E}\left[\sup_{t \in [0,1]} \left\|\dot{X}_t\right\|\right] < \infty,$$

*where $\mu_t$ denotes the distribution of $X_t \in \mathcal{H}$ for each $t \in [0, 1]$.*

With these notions, we can now define a rectified flow.

**Definition 4.** *We say that a stochastic process $\mathbb{X} = \{X_t\}_{t\in[0,1]}$ is **rectifiable** if Assumption 2 holds for $v^{\mathbb{X}}(t, x)$ and Assumption 3 holds for $\mathbb{X}$. In this case, the process defined by*

$$Z_t = Z_0 + \int_0^t v^{\mathbb{X}}(s, Z_s)ds, \quad Z_0 \sim X_0, \tag{2}$$

*is called the **rectified flow** induced by $\mathbb{X}$.*

Unlike diffusion models, which rely on stochastic differential equations (SDEs), rectified flows are defined in terms of a deterministic ordinary differential equation (ODE) guided by $v^{\mathbb{X}}(t, x)$.

### 3.2 THE MARGINAL PRESERVING PROPERTY

A key property of a rectified flow is that it preserves the marginal distributions of the original stochastic process. This is formalized in the following theorem which generalizes Liu et al. (2022) from a finite-dimensional Euclidean space to a separable Hilbert space $\mathcal{H}$.

**Theorem 5.** *Assume that an $\mathcal{H}$-valued stochastic process $\{X_t\}$ is rectifiable, and let $\{Z_t\}$ be the induced rectified flow. Then, for all $t \in [0, 1]$, $Z_t \stackrel{d}{=} X_t$.*

The central challenge in proving the theorem lies in developing the superposition principle within a general separable Hilbert space setting. Informally, the superposition principle says that any distribution solving the "continuity equation" driven by $v^{\mathbb{X}}$ can be "disintegrated" into a distribution over paths that solve the associated ODE (2). The proof requires establishing several nontrivial technical results and rigorously connecting the resulting measure-theoretic decomposition to the solution of Equation (1). We defer the exact statement of the superposition principle and the proof to Section A in the appendix. In Appendix Section E, we empirically verify Theorem 5 on synthetic data where $X_0$ and $X_1$ are sampled from tractable distributions and $v^{\mathbb{X}}$ can be computed in closed form.

Note that $Z_t$ derives its randomness entirely from the random initial condition $Z_0$. Intuitively, this result states that while $Z_t$ evolves deterministically given $Z_0$ according to an ODE, it preserves the marginal distributions as $X_t$. When $X_t$ is constructed to interpolate between two distributions, where $X_0$ corresponds to a simple reference distribution (e.g., i.i.d. Gaussian noise) and $X_1$ represents the target data distribution, then learning the velocity field $v^{\mathbb{X}}$ enables sampling from the data distribution by solving (2) with randomly initialized $Z_0$.

The rectified flow method of Liu et al. (2022) corresponds to a specific choice of $\mathbb{X} = \{X_t\}$ given by

$$X_t = tX_1 + (1 - t)X_0, \tag{3}$$

where $X_0$ and $X_1$ are independently sampled from the noise and data distributions, respectively. This formulation was shown to be effective in finite-dimensional Euclidean spaces.

If we define $X_t = tX_1 + (1-t)X_0$ as in (3), where $X_0, X_1 \in \mathcal{H}$, then $\dot{X}_t = X_1 - X_0$. The velocity field $v^{\mathbb{X}}(t, x)$, modeled by a neural network $v_\theta$, can be trained to minimize

$$\min_\theta \int_0^1 \mathbb{E}_{x\sim\mathbb{X}}\|(x_1 - x_0) - v_\theta(x_t, t)\|^2 dt, \tag{4}$$

where $x$ is a realization of the stochastic process and $x_t$ is the value of $x$ at time $t$. Note that the objective can be approximated by randomly sampled data $X_1$, noise $X_0$, and timestamp $t$.

For sampling, we numerically integrate the ODE: $Z_t = Z_0 + \int_0^t v_\theta(s, Z_s)ds, \quad Z_0 \sim X_0$, allowing us to generate new samples from $X_1$ by evolving from $X_0$ using the learned expected velocity $v_\theta$. This approach is computationally more efficient than stochastic sampling methods used in diffusion models, as it avoids the nuances of reversing an SDE.

### 3.3 PROPERTIES OF RECTIFIED FLOW

The desirable properties of rectified flows (3) in finite-dimensional settings, as discussed in Liu et al. (2022)—including transport cost reduction and the straightening effect—also hold in the infinite-dimensional Hilbert space setting. These results follow directly from the marginal-preserving property established in Theorem 5, using arguments that largely mirror those in finite dimensions. As the extensions mainly involve technical but routine bookkeeping, we defer the detailed statements and proofs to Section B in the appendix.

## 4 CONNECTIONS TO OTHER FUNCTIONAL MODELS

We place our work in the broader context of functional generative modeling by examining its relationship to functional flow matching (Kerrigan et al., 2024) and functional probability flow ODE (Na et al., 2025). Our framework accommodates interpolation paths of the form:

$$X_t = \alpha_t X_1 + \beta_t X_0, \tag{5}$$

where $\alpha_t, \beta_t$ are continuously differentiable functions of time $t$. This generalized parameterization recovers rectified flow when $\alpha_t = t$, $\beta_t = 1 - t$. We call Equation (5) the nonlinear extension of the function rectified flow. We show that both functional flow matching (Kerrigan et al., 2024) and functional probability flow ODE (Na et al., 2025) can be viewed as nonlinear extensions of functional rectified flow within our general framework.

To facilitate our discussion, we first define the Cameron–Martin space and the notion of a Wiener process in Hilbert space. Let $Q$ be a trace-class, positive, symmetric operator on $\mathcal{H}$. The associated Cameron–Martin space is given by $\mathcal{H}_Q := Q^{\frac{1}{2}}(\mathcal{H})$, the range of $Q^{\frac{1}{2}}$, equipped with the inner product $\langle x, y \rangle_{\mathcal{H}_Q} := \left\langle Q^{-\frac{1}{2}} x, Q^{-\frac{1}{2}} y \right\rangle_{\mathcal{H}}$. A $Q$-Wiener process $W_t$ is a continuous Gaussian process in $\mathcal{H}$ satisfying $W_0 = 0$ and having independent, stationary increments, with $W_t - W_s \sim \mathcal{N}(0, (t - s)Q)$ for all $0 \le s \le t$.

### 4.1 COMPARISON WITH FUNCTIONAL FLOW MATCHING

By examining Equation (6) in Kerrigan et al. (2024), we can directly see that Equation (5) subsumes their proposed flow matching models as special cases. In Table 1, we summarize how the "VP" and "OT" paths introduced in their work arise as specific instances of $(\alpha_t, \beta_t)$.

The framework of Kerrigan et al. (2024) relies on a strong measure-theoretic assumption that may not hold even in finite-dimensional Euclidean space. Suppose there exists a stochastic process $\{X_t\}_{t \in [0,1]}$ interpolating $X_0$ (noise) and $X_1$ (data), with $\mu_t$ denoting the marginal law of $X_t$, and $\mu_t^x$ the conditional law of $X_t$ given $X_1 = x$. Kerrigan et al. (2024) assume: $\mu_t^x \ll \mu_t$, *i.e.*, the conditional measure $\mu_t^x$ must be absolutely continuous with respect to the marginal $\mu_t$ for almost every $x$. As admitted by the authors in Kerrigan et al. (2024), this assumption is **difficult to satisfy and verify, and generally fails even in finite-dimensional Euclidean space**. If $X_0$ is a trace-class Gaussian process, this reduces to the requirement that $X_1$ lies in the Cameron–Martin space of the initial Gaussian measure, i.e., $X_1 \in C_0^{1/2}(\mathcal{H})$ when $X_0 \sim \mathcal{N}(0, C_0)$. As also noted in Kerrigan et al. (2024), even this condition is **highly restrictive and challenging to verify in practice**. In section D in the Appendix, we show that this assumption would fail in the trivial case where $X_1$ and $X_0$ are gaussian and equal in distribution. Although we assume that the processes are pathwise continuously differentiable, this assumption reflects a design choice rather than a data assumption.

Table 1: Functional flow matching paths proposed in Kerrigan et al. (2024) are special cases of (5)

| Flow Path | $\alpha_t$ | $\beta_t$ |
| --- | --- | --- |
| OT (Optimal Transport) | $t$ | $1 - (1 - \sigma_{\min})t, \sigma_{\min} \in (0, 1)$ |
| VP (Variance Preserving) | arbitrary $\in [0, 1]$ | $\sqrt{1 - \alpha_t^2}$ |

## 4.2 COMPARISON WITH FUNCTIONAL PROBABILITY FLOW ODE

Throughout this section, we adopt the convention that time $t = 0$ corresponds to the data distribution and $t = 1$ to the noise distribution. This is the reverse of the convention used in the rest of the paper, and is chosen to align with the standard formulation of score-based generative modeling and SDEs. This section also references technical concepts (e.g., $Q$-Wiener process, Cameron-Martin space) that will be defined in the appendix.

The concurrent work of Na et al. (2025) introduces an analogue of the probability flow ODE from Song et al. (2021), adapted to infinite-dimensional Hilbert spaces. Specifically, they consider the following variance-preserving SDE in $\mathcal{H}$:

$$dY_t = -\frac{\sigma_t}{2} Y_t \, dt + \sqrt{\sigma_t} \, dW_t, \quad Y_0 \sim P_{\text{data}},$$

where $W_t$ is a $Q$-Wiener process in $\mathcal{H}$, $\sigma_t$ is a bounded, continuous function taking values on $\mathbb{R}_{\geq 0}$. Let $\{\zeta_i\}$ denote an orthonormal basis of $\mathcal{H}$ consisting of eigenvectors of $Q$, and let $\mathcal{H}_Q$ denote the corresponding Cameron–Martin space. To support the analysis, we recall the formal definition of the logarithmic gradient.

**Definition 6.** *A Borel probability measure $\mu$ is said to be Fomin differentiable along $h \in \mathcal{H}_Q$ if there exists a function $\rho_h^\mu \in L^1(\mathcal{H}, \mu)$ such that for all cylinder functions of the form*

$$F(x) = f\left(\langle \zeta_1, x \rangle, \ldots, \langle \zeta_m, x \rangle\right), \quad f \in \mathcal{C}_0^\infty(\mathbb{R}^m), \, m \in \mathbb{N},$$

*we have*

$$\int_{\mathcal{H}} \partial_h F(x) \, \mu(dx) = -\int_{\mathcal{H}} F(x) \, \rho_h^\mu(x) \, \mu(dx),$$

*where $\partial_h$ denotes the Gâteaux differential along $h$. Let $\mathcal{K}$ be a subspace of $\mathcal{H}$. If there exists a function $\rho_{\mathcal{K}}^\mu : \mathcal{H} \to \mathcal{H}$ such that*

$$\langle \rho_{\mathcal{K}}^\mu(x), h \rangle_{\mathcal{K}} = \rho_h^\mu(x) \quad \text{for all } x \in \mathcal{H}, \, h \in \mathcal{K},$$

*then $\rho_{\mathcal{K}}^\mu$ is called the logarithmic gradient of $\mu$ along $\mathcal{K}$.*

During training, a score network $S(t, Y_t)$ is learned to approximate the logarithmic gradient by minimizing the objective

$$\int_0^1 \mathbb{E}_{Y_0 \sim P_{\text{data}}} \mathbb{E}_{Y_t \sim \mu_{t|Y_0}} \left\| S(t, Y_t) - \rho_{\mathcal{H}_Q}^{\mu_{t|Y_0}}(Y_t) \right\| dt,$$

where $\mu_{t|Y_0}$ denotes the conditional distribution of $Y_t$ given $Y_0$. Let $S^*$ denote its minimizer.

For sampling, Na et al. (2025) propose the following probability flow ODE, to be solved in reverse time from $t = 1$ to $t = 0$:

$$dY_t = -\frac{\sigma_t}{2}\left(Y_t + S^*(t, Y_t)\right) dt, \quad Y_1 \sim \mathcal{N}(0, Q). \tag{6}$$

The following proposition shows how this ODE fits within our rectified flow framework.

**Proposition 7.** *Equation (6) is the rectified flow induced by the process $Y_t'$ defined as*

$$Y_t' = \eta(t) \, Y_0' + \sqrt{\kappa(t)} \, U,$$

*where $\eta(t) = \exp\left(-\frac{1}{2}\int_0^t \sigma_s \, ds\right)$, $\kappa(t) = \int_0^t \exp\left(-\int_s^t \sigma_\tau d\tau\right) \sigma_s \, ds$, and $Y_0' \sim P_{data}$, $U \sim \mathcal{N}(0, Q)$ are sampled independently. Equivalently, $\mathbb{E}[\dot{Y}_t' \mid Y_t'] = -\frac{\sigma_t}{2}\left(Y_t' + S^*(t, Y_t')\right).$*

Thus, the probability flow ODE (6) corresponds to the time-reversal of the nonlinear rectified flow in Equation (5), with the interpolation weights given by $\alpha_t = \eta(1 - t)$ and $\beta_t = \sqrt{\kappa(1 - t)}$.

## 5 ARCHITECTURES FOR FUNCTIONAL GENERATIVE MODELING

Let $\mathcal{H}$ be a Hilbert space of functions from some coordinate space $M$ to $\mathbb{R}$. To implement functional rectified flows in practice, one must approximate the expected velocity field $v^{\mathbb{X}}(x_t, t) : \mathcal{H} \times [0, T] \rightarrow \mathcal{H}$, which is inherently defined over an infinite-dimensional domain. Directly learning such a mapping is generally intractable. However, if $\mathcal{H} = L_2(M)$, under the conditions of Theorem 2 in Franzese et al. (2023), any $x \in \mathcal{H}$ can be fully characterized by its pointwise evaluations $\{(x[p_i], p_i)\}$, where $p_i \in M$. This insight enables the design of networks that act on sampled representations rather than abstract functional inputs, thus making implementation feasible. Following Franzese et al. (2023) and Kerrigan et al. (2024), we consider three practical architectures for this purpose: *Implicit Neural Representations* (INRs), *Transformers*, and *Neural Operators*. We include this discussion of architectural choices for completeness but refer the reader to Franzese et al. (2023) for a comprehensive treatment.

The first approach builds on *implicit neural representations* (INRs) Sitzmann et al. (2020) and the model-agnostic meta-learning (MAML) paradigm Finn et al. (2017). Following the modulation strategy introduced for functional diffusion processes by Franzese et al. (2023), we represent the velocity field $v_\theta(x_t, t)$ with a network $n(\psi, t, \theta) : M \rightarrow \mathbb{R}$ where $\theta$ is the shared base parameter and $\psi$ is a *sample-specific* modulation vector that is adapted online to encode each $x_t$. Formally,

$$v_\theta(x_t, t) = n(\psi^*, t, \theta), \quad \psi^* = \arg\min_\psi \sum_{p_i} (n(\psi, t, \theta)[p_i] - x_t[p_i])^2. \tag{7}$$

For every $x_t$ we initialize $\psi = \mathbf{0}$ and perform a small number of gradient-descent steps, with $\theta$ fixed, to minimize the residual in (7). During training, the global parameters $\theta$ are updated via the loss (4), whereas $\psi$ is re-optimized from scratch for each $x_t$, enabling the network to have a functional representation while operating on finite discretizations $\{x_t[p_i], p_i\}$.

Our second approach employs *transformer architectures* (Vaswani et al., 2017), treating discretized function evaluations as sequences with positional information. For each $x_t$, we consider the set of finite discretizations $\{x_t[p_i]\}$ as input features and their corresponding coordinates $\{p_i\}$ as positions. Each $x_t[p_i]$ is embedded into a higher-dimensional vector space, and summed with the positional encodings of $p_i$, resulting in a sequence of embeddings $\{y_i\}$. This sequence is then processed by a transformer to obtain $\{v_\theta(x_t, t)[p_i]\}$. We note that Cao (2021) interpret transformers as mappings between Hilbert spaces, and Kovachki et al. (2021) discuss the learned rather than guaranteed nature of resolution invariance of transformers. While these theoretical perspectives are relevant, we do not explore them further and refer interested readers to the original papers for a detailed discussion.

Neural operators (Kovachki et al., 2021; Kossaifi et al., 2024) offer a principled framework for modeling functions in infinite-dimensional settings and have demonstrated success in various applications (Peng et al., 2024; Qin et al., 2025). A neural operator learns a mapping $\mathcal{G} : \mathcal{H} \rightarrow \mathcal{H}$, trained using finite evaluations of functions. Once trained, it maps a uniform discretization $\{(x[p_i], p_i)\}$ to output values at the same locations, producing $\{\mathcal{G}(x)[p_i]\}$. In our implementation, we directly parameterize the velocity field $v_\theta$ using a neural operator. Similar to the transformer-based approach, the positions $\{p_i\}$ are encoded via positional embeddings and appended to the embeddings of the corresponding $\{x_t[p_i]\}$. Although neural operators typically require inputs to be defined on fixed, uniformly spaced grids and produce outputs at the same grid locations—limiting flexibility for irregular or adaptive sampling—they have been shown to be highly effective for PDE data (Kerrigan et al., 2024). Bond-Taylor and Willcocks (2024) implement an infinite-dimensional generative model using a neural operator-based architecture but rely on a specialized diffusion autoencoder (Preechakul et al., 2022), making their approach highly specialized and not applicable to our setting.

## 6 EXPERIMENTS

We evaluate the proposed Functional Rectified Flow (FRF) model on three datasets—MNIST (MIT License) (LeCun et al., 2010), CelebA $64 \times 64$ (CC BY-SA 4.0) (Liu et al., 2015), and the Navier–Stokes dataset (MIT License) (Li et al., 2022), using INR, transformer, and neural operator implementations, respectively. Our aim is to assess whether extending rectified flow to infinite-dimensional Hilbert spaces yields competitive or superior performance compared with state-of-the-art functional generative models across different architectural classes. Importantly, in Section 6.1.1 (INR), Section

6.1.2 (ViT), and Section 6.2 (Neural Operator), we use **exactly the same network architectures as FDP and FFM, adopting the architectures from their original papers to avoid bias**. Thus, the performance gains observed in our experiments arise solely from the proposed functional flow matching objective, not from increased model capacity. For clarity, we present results in two parts: (1) image-generation experiments on MNIST and CelebA, where each image is considered as a function mapping coordinates in $[0, 1]^2$ to pixel values in $\mathbb{R}$ and $\mathbb{R}^3$, respectively, and (2) PDE experiments on the Navier–Stokes dataset. Additional experimental details appear in Section C of the appendix.

## 6.1 IMAGE DATASETS

For image datasets, we focus our comparisons on Functional Diffusion Processes (FDP) (Franzese et al., 2023) and $\infty$-DIFF (Bond-Taylor and Willcocks, 2024), which are among the most competitive methods for image generation in functional settings. To ensure a fair comparison, we adopt the exact same architectural design as Franzese et al. (2023), without introducing any specialized enhancements. FRF is implemented in JAX (Bradbury et al., 2018), building directly on the released codebase of Franzese et al. (2023) (Apache-2.0 License). For INR-based models, modulation codes are optimized using 3 steps of gradient descent per sample. For both MNIST and CelebA experiments, $X_0$ is sampled from the stationary distribution of the functional SDE (9) in Franzese et al. (2023).

To evaluate generative performance, we consider the Fréchet Inception Distance (FID) (Heusel et al., 2017) and the FID-CLIP score (Kynkäänniemi et al., 2023), where lower values indicate higher visual fidelity and closer alignment with the target data distribution.

### 6.1.1 INR ON MNIST

On MNIST ($32 \times 32$), we benchmark an INR-based FRF model against the Functional Diffusion Process (FDP) (Franzese et al., 2023). We specifically adopt the INR architecture to test the effectiveness of FRF on lightweight models, as INRs naturally have a small parameter footprint while retaining strong functional approximation capabilities.

As shown in Table 2, our INR-based FRF achieves a lower FID score than FDP while using the same number of parameters, indicating that functional rectified flow leads to improved sample quality even with lightweight models.

Figure 1 further illustrates the qualitative advantages of FRF. Notably, the super-resolved samples produced by our model at both $64 \times 64$ and $128 \times 128$ resolutions (Figures (b) and (d)) exhibit smoother digit contours compared to the naïvely upscaled real MNIST images (Figures (c) and (e)). This highlights the model's ability to generate coherent high-resolution samples by leveraging its continuous functional representation while being trained on a lower resolution.

Table 2: Results on MNIST ($32 \times 32$) using INR.

| Method | FID ($\downarrow$) | Params |
|---|---|---|
| FRF (INR, ours) | **0.41** | $\approx 0.1$M |
| FDP (INR) | 0.43 | $\approx 0.1$M |

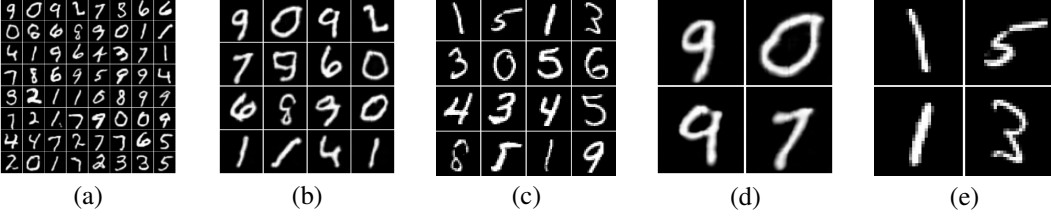

|  |  |  |  |  |
|---|---|---|---|---|
| (a) | (b) | (c) | (d) | (e) |

Figure 1: Qualitative results on MNIST: (a) samples generated at the original $32 \times 32$ resolution; (b) super-resolved samples at $64 \times 64$; (c) real MNIST images upscaled to match (b); (d) super-resolved samples at $128 \times 128$; (e) real MNIST images upscaled to match (d).

### 6.1.2 TRANSFORMER ON CELEBA

On CelebA ($64 \times 64$), we evaluate transformer-based FRF models and compare against FDP (both INR and transformer variants) (Franzese et al., 2023), FD2F (Dupont et al., 2022), and $\infty$-DIFF (Bond-Taylor and Willcocks, 2024). We specifically adopt the transformer architecture to demonstrate that FRF is capable of generating high-quality and visually compelling images on complex datasets.

Table 3 shows that FRF with a transformer backbone outperforms other functional generative models in terms of both FID and FID-CLIP scores, while also being significantly more parameter-efficient than $\infty$-DIFF. Figure 2 further illustrates the visual quality of samples generated by FRF. The model successfully captures rich facial details and produces sharp, high-fidelity images.

Table 3: Results on CelebA ($64 \times 64$) using transformer-based architectures.

| Method | FID ($\downarrow$) | FID-CLIP ($\downarrow$) | Params |
|---|---|---|---|
| FRF (Vision Transformer, ours) | **6.63** | **3.70** | $\approx$ 20M |
| FDP (INR) | 35.00 | 12.44 | $\approx$ 1M |
| FDP (Vision Transformer) | 11.00 | 6.55 | $\approx$ 20M |
| FD2F | 40.40 | – | $\approx$ 10M |
| $\infty$-DIFF | – | 4.57 | $\approx$ 100M |

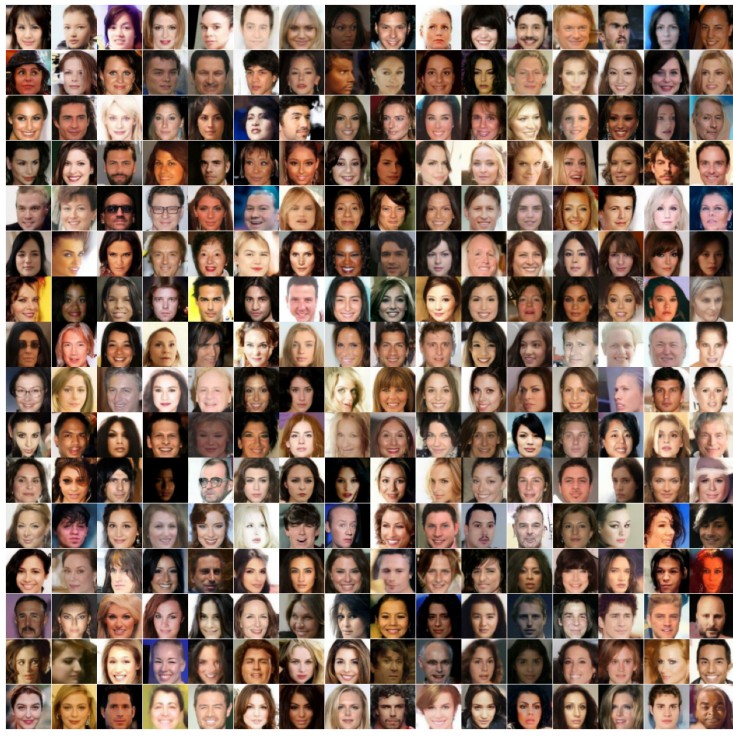

Figure 2: Qualitative results of functional rectified flow with vision transformer.

## 6.2 NEURAL OPERATOR ON NAVIER-STOKES DATASET

We evaluate the proposed FRF model with a neural operator backbone on the Navier-Stokes dataset, a benchmark consisting of solutions to the Navier-Stokes equations on a 2D torus. This dataset captures complex fluid dynamics, making it a challenging testbed for functional generative models. To be consistent with prior work (Kerrigan et al., 2024), we use a $\frac{1}{2}$-Matern kernel to sample the initial condition $X_0$. Our code is built upon the open-sourced codebase of Kerrigan et al. (2024) (MIT license).

We compare our model against several baselines: the Denoising Diffusion Operator (DDO) (Lim et al., 2023a) using the NCSN noise scale, GANO (Rahman et al., 2022), Functional DDPM (Kerrigan et al., 2023), and Functional Flow Matching (FFM) (Kerrigan et al., 2024).

Following Kerrigan et al. (2024), we evaluate the model performance by *Density MSE*, which is computed by applying pointwise Kernel Density Estimation (KDE) to 1,000 generated and 1,000 real samples, followed by computing the Mean Squared Error between the resulting estimated densities. This metric evaluates how well the generated samples reproduce the overall spatial distribution of the data.

As shown in Table 4, the FRF model achieves the lowest density MSE among all methods, indicating that the generated samples accurately match the distribution of real samples in the spatial domain.

Table 4: MSEs between the density of the real and generated samples of the Navier-Stokes dataset.

| Method | Density MSE (Mean $\pm$ Std) |
| --- | --- |
| FRF | $\mathbf{2.39 \times 10^{-5}} \ \pm \ \mathbf{4.45 \times 10^{-6}}$ |
| FFM | $4.50 \times 10^{-5} \ \pm \ 1.52 \times 10^{-5}$ |
| DDPM | $1.02 \times 10^{-4} \ \pm \ 8.20 \times 10^{-6}$ |
| DDO | $9.61 \times 10^{-3} \ \pm \ 1.26 \times 10^{-2}$ |
| GANO | $4.16 \times 10^{-3} \ \pm \ 1.82 \times 10^{-3}$ |

# 7 CONCLUSION, LIMITATIONS, BROADER IMPACT, AND FUTURE DIRECTIONS

We have introduced a functional extension of rectified flow by lifting it to general Hilbert spaces. Our theoretical results demonstrate that the marginal preserving property is preserved in the infinite-dimensional setting. This provides a principled and tractable foundation for functional generative modeling. We demonstrate experimentally that our method achieves superior performance compared to existing functional generative models. Notably, while most competing approaches develop specialized architectures tailored to their methods, we demonstrate the flexibility of our framework by applying it across three distinct model architectures—each originally designed for one of our competitors. Although our framework is broadly applicable, domain-specific architectures and inductive biases may still be required for optimal performance in high-complexity tasks.

As with all generative models, risks such as misuse in synthetic media generation remain, including the potential for creating misleading or harmful content. We encourage responsible use and further research into interpretability, robustness, and safety in functional generative models to mitigate these concerns. At the same time, functional generative models offer significant positive societal impacts, including advancing scientific discovery through improved simulation of complex physical systems and supporting creative industries with new tools for high-quality content generation.

A natural theoretical direction is to extend the rectified flow framework to general Banach spaces. The key obstacle is the reliance on orthonormal bases in the current analysis, which are available in Hilbert spaces but absent in generic Banach spaces. On the empirical side, an important extension is to develop functional versions of generative models that operate on a broader class of data modalities, such as irregularly sampled time series and 3D geometric objects.

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

## A   PROOFS AND REMARKS

In this section, we present the proofs together with remarks on technical details.

### A.1   REMARKS ON THE EXISTENCE OF THE VELOCITY FIELD

Under our setting, the existence of $v^{\mathbb{X}}$ follows from standard results.

**Proposition 8** (Proposition 1.10 of Da Prato and Zabczyk (1992)). *Assume that $E$ is a separable Banach space. Let $V$ be a Bochner integrable $E$-valued random variable defined on $(\Omega, \mathcal{S}, \mathbb{P})$ and let $\mathcal{G}$ be a $\sigma$-field contained in $\mathcal{S}$. There exists a unique, up to a set of $\mathbb{P}$-probability zero, integrable $E$-valued random variable $R$, measurable with respect to $\mathcal{G}$ such that*

$$\int_A V \, d\mathbb{P} = \int_A R \, d\mathbb{P}, \quad \forall A \in \mathcal{G}.$$

*The random variable $R$ is denoted as $\mathbb{E}[V \mid \mathcal{G}]$ and called the conditional expectation of $V$ given $\mathcal{G}$.*

In our case, the state space $\mathcal{H}$ is a separable Banach space. Hence, it suffices to assume $\mathbb{E}\|\dot{X}_t\| < \infty$ for integrability. Under this condition, $\mathbb{E}[\dot{X}_t \mid \sigma(X_t)]$ exists and is $\sigma(X_t)$-measurable. By the Doob–Dynkin lemma, there exists a measurable function $v_t : \mathcal{H} \to \mathcal{H}$ such that

$$\mathbb{E}[\dot{X}_t \mid \sigma(X_t)] = v_t(X_t).$$

Defining $v^{\mathbb{X}}(t, x) = v_t(x)$ for $x \in \text{supp}(X_t)$ and $v^{\mathbb{X}}(t, x) = 0$ otherwise (which affects only a set of measure zero) yields the desired vector field. For completeness, we recall the Doob–Dynkin lemma:

**Lemma 9** (Lemma 1.14 of Kallenberg (2021)). *Let $f, g$ be measurable functions from $(\Omega, \mathcal{A})$ into measurable spaces $(S, \mathcal{S})$ and $(T, \mathcal{T})$, where $S$ is Borel. The following are equivalent:*

1. *$f$ is $g$-measurable, i.e. $\sigma(f) \subset \sigma(g)$,*

2. *there exists a measurable mapping $h : T \to S$ such that $f = h \circ g$.*

This shows that the velocity field $v^{\mathbb{X}}$ is well-defined under mild integrability conditions.

### A.2   DEFINITIONS AND TECHNICAL LEMMA

In this section, we present definitions and technical lemmas for the invocation of the superposition principle in Hilbert space. We first review Fréchet differentiability on a normed space.

**Definition 10.** *Let $E$ and $F$ be normed vector spaces, $O$ an open subset of $E$ containing $0$. The derivative of $f : E \to F$ at $x \in O$ is a continuous linear map $D_x f$ from $E$ to $F$ such that*

$$\lim_{h \to 0} \frac{f(x + h) - f(x) - D_x f(h)}{\|h\|} = 0$$

When $E = \mathcal{H}$ and $F = \mathbb{R}$, $D_x f \in \mathcal{H}^*$ where $\mathcal{H}^*$ is the dual space of $\mathcal{H}$. By the Riesz representation theorem, there is a unique $g \in \mathcal{H}$ such that $D_x f(\cdot) = \langle \cdot, g \rangle$, and we denote such $g$ by $\nabla f(x)$. We define the second-order derivative $\nabla^2 f(x)$ as the derivative of $x \in \mathcal{H} \to \nabla f(x) \in \mathcal{H}$.

**Definition 11.** *Let $\mathcal{H}$ be a separable Hilbert space with orthonormal basis $\{e_i\}$ and $\varphi : \mathcal{H} \to \mathbb{R}$. Define its asymptotic Lipschitz constant $L_\varphi(x)$ at $x \in \mathcal{H}$ by*

$$L_\varphi(x) = \lim_{\epsilon \to 0^+} L_\varphi(x, \epsilon),$$

*where*

$$L_\varphi(x, \epsilon) = \sup_{u, v \in B_\epsilon(x)} \frac{|\varphi(u) - \varphi(v)|}{\|u - v\|}.$$

Notice that $L_\varphi$ is upper semicontinuous (Stepanov and Trevisan, 2017).

**Lemma 12.** *If $\varphi : \mathcal{H} \to \mathbb{R}$ is a twice-differentiable function with bounded second-order derivative, then $L_\varphi(x) = \|\nabla \varphi(x)\|$.*

*Proof.* Fix $\epsilon > 0$ and $x \in \mathcal{H}$. We first show $L_\varphi(x) \leq \|\nabla\varphi(x)\|$.

Let $a, b \in B_\epsilon(x)$ and $b \neq a$. By the mean value theorem (Theorem 3.2 of Coleman (2012)),

$$\varphi(b) - \varphi(a) = \langle \nabla\varphi(c), b - a \rangle$$
$$= \langle \nabla\varphi(x), b - a \rangle + \langle \nabla\varphi(c) - \nabla\varphi(x), b - a \rangle$$

where $c = ta + (1-t)b$ for some $t \in [0, 1]$. Hence,

$$|\varphi(b) - \varphi(a)| \leq \|\nabla\varphi(x)\|\|b - a\| + \|\nabla\varphi(c) - \nabla\varphi(x)\|\|b - a\|$$
$$\leq \|\nabla\varphi(x)\|\|b - a\| + \sup_{c' \in \{t'x + (1-t')c : t' \in [0,1]\}} \|\nabla^2\varphi(c')\|_{\mathcal{H},\mathcal{H}} \|b - a\|\|c' - x\|$$

We obtain the second inequality by applying the mean value inequality (Corollary 3.2 of Coleman (2012)) to $\nabla\varphi$. Suppose $\nabla^2\varphi$ is bounded by a constant $B_0$. Note that both $c, c' \in B_\epsilon(x)$. Hence,

$$\frac{|\varphi(b) - \varphi(a)|}{\|b - a\|} \leq \|\nabla\varphi(x)\| + B_0\epsilon$$

The choice of $a, b$ is arbitrary, so $L_\varphi(x, \epsilon) \leq \|\nabla\varphi(x)\| + B_0\epsilon$ and $L_\varphi(x) \leq \|\nabla\varphi(x)\|$.

Now it remains to show $L_\varphi(x) \geq \|\nabla\varphi(x)\|$. WOLOG, we assume $\nabla\varphi(x) \neq 0$ and let $v = \frac{\nabla\varphi(x)}{\|\nabla\varphi(x)\|}$. Let $y_n = x + \frac{1}{n}v, z_n = x, n \in \mathbb{N}$.

Fix $\alpha > 0$. Then $\exists N \in \mathbb{N}$ such that $\forall n \geq N, y_n \in B_\epsilon(x)$ and by definition of Fréchet differentiability

$$\left| \varphi(x + \frac{1}{n}v) - \varphi(x) - \left\langle \nabla\varphi(x), \frac{1}{n}v \right\rangle \right| \leq \alpha \left\| \frac{v}{n} \right\| = \frac{\alpha}{n}.$$

Hence, $\varphi(x + \frac{1}{n}v) - \varphi(x) \geq \langle \nabla\varphi(x), \frac{1}{n}v \rangle - \frac{\alpha}{n} = \frac{1}{n}\|\nabla\varphi(x)\| - \frac{\alpha}{n}$.

For $y_n \in B_\epsilon(x)$,

$$L_\varphi(x, \epsilon) \geq \frac{|\varphi(y_n) - \varphi(z_n)|}{\|y_n - z_n\|}$$
$$= \frac{|\varphi(x + \frac{1}{n}v) - \varphi(x)|}{\frac{1}{n}}$$
$$\geq \frac{\varphi(x + \frac{1}{n}v) - \varphi(x)}{\frac{1}{n}}$$
$$\geq \|\nabla\varphi(x)\| - \alpha.$$

Since $\alpha$ is arbitrary, $L_\varphi(x, \epsilon) \geq \|\nabla\varphi(x)\|$. Therefore, $L_\varphi(x) = \|\nabla\varphi(x)\|$. $\qquad\square$

**Definition 13.** *Let $\mathcal{A}$ be the cylindrical functions on $\mathcal{H}$, i.e.,*

$$\mathcal{A} = \left\{ \varphi \in \mathcal{H} \to \mathbb{R} : \varphi = \phi \circ P_d, d \text{ is a non-negative integer}, \phi \in C^\infty(\mathbb{R}^d) \text{ has compact support} \right\}$$

*where $P_d(x) = (\langle x, e_1 \rangle, \ldots, \langle x, e_d \rangle)$ is a finite dimensional projection and $C^\infty(\mathbb{R}^d)$ is the set of compactly supported function from $\mathbb{R}^d$ to $\mathbb{R}$.*

$\mathcal{A}$ is an algebra (*i.e.*, a set closed under pointwise linear combinations and products), and a constant function is trivially an element in $\mathcal{A}$ by letting $d = 0$. By Lemma 12 $L_\varphi(x) = \|\nabla\varphi(x)\|, \forall\varphi \in \mathcal{A}$. Let $v_t : \mathcal{H} \to \mathcal{H}$, for $t \in [0, 1]$ be a vector field. $v_t$ defines an operator $V_t : \mathcal{A} \to \mathbb{R}$ acting on elements in $\mathcal{A}$, *i.e.*, for $\varphi \in \mathcal{A}$, $(V_t\varphi)(x) = \langle \nabla\varphi(x), v_t(x) \rangle$. Hence, $V_t$ satisfies $V_t(fg) = fV_t(g) + gV_t(f)$. Also, $|(V_t\varphi)(x)| = |\langle \nabla\varphi(x), v_t(x) \rangle| \leq \|v_t(x)\|L_\varphi(x)$.

We have to show one more lemma to invoke the results of Stepanov and Trevisan (2017).

**Lemma 14.** *Let $\varphi$ be a Lipschitz, bounded function on $\mathcal{H} \times [0, 1]$, with metric $((x, t), (y, s)) \to (\|x - y\|^2 + |t - s|^2)^{\frac{1}{2}}$. Then there is a sequence of uniformly bounded functions $\{\varphi_k\}$ on $\mathcal{H} \times [0, 1]$ that are also Lipschitz with uniform Lipschitz constants, such that $\varphi_k(\cdot, t) \in \mathcal{A}$ for $t \in [0, 1]$, $\lim_k \varphi_k = \varphi$ pointwisely, and*

$$\limsup_k L_{\varphi_k(\cdot, t)}(x) \leq L_{\varphi(\cdot, t)}(x), \forall(x, t) \in \mathcal{H} \times [0, 1].$$

*Proof.* Let $\{e_k\}$ be an orthonormal basis of $\mathcal{H}$. Let $C_\varphi$ be the Lipschitz constant of $\varphi$.

For an integer $k > 0$, recall that $P_k : \mathcal{H} \to \mathbb{R}^k$, $P_k(x) = (\langle x, e_1 \rangle, \langle x, e_2 \rangle, \ldots, \langle x, e_k \rangle)$. Define $\pi_k(x) : \mathcal{H} \to \mathcal{H}$, $\pi_k(x) = \sum_{i=1}^k \langle x, e_i \rangle e_i$ and $\psi_k : \mathbb{R}^k \times [0, 1] \to \mathbb{R}$, $\psi_k(r, t) = \varphi(\sum_{i=1}^k r_i e_i, t)$. Note that $\|P_k(x) - P_k(y)\|_{\mathbb{R}^k} = \|\pi_k(x) - \pi_k(y)\|_{\mathcal{H}} \leq \|x - y\|_{\mathcal{H}}$, and $\psi_k(P_k(x), t) = \varphi(\pi_k(x), t)$.

Also note that

$$|\psi_k(r, t) - \psi_k(\bar{r}, s)| \leq C_\varphi \left( \left\| \sum_{i=1}^k r_i e_i - \sum_{i=1}^k \bar{r}_i e_i \right\|_{\mathcal{H}}^2 + (t - s)^2 \right)^{1/2}$$

$$= C_\varphi \left( \|r - \bar{r}\|_{\mathbb{R}^k}^2 + (t - s)^2 \right)^{1/2}.$$

So $\psi_k$ is also $C_\varphi$-Lipschitz.

Fix an integer $k > 0$. We first approximate $\psi_k$ using a standard mollification approach. Following Appendix C.5 of Evans (2010), we define the standard mollifier function for $\epsilon > 0$:

$$\eta_\epsilon : \mathbb{R}^k \to \mathbb{R}, \eta_\epsilon(r) = \frac{1}{\epsilon^k} \eta(\frac{r}{\epsilon}),$$

where $\eta : \mathbb{R}^k \to \mathbb{R} \in C^\infty(\mathbb{R}^k)$,

$$\eta(r) := \begin{cases} C \exp\left( \frac{1}{|r|^2 - 1} \right) & \text{if } \|r\|_{\mathbb{R}^k} < 1 \\ 0 & \text{if } \|r\|_{\mathbb{R}^k} \geq 1 \end{cases},$$

and the constant $C > 0$ is selected so that $\int_{\mathbb{R}^k} \eta = 1$. Let $B_\epsilon^k(r)$ be the $\epsilon$-ball in $\mathbb{R}^k$ around $r$ and note that $\eta_\epsilon$ is supported on $B_\epsilon^k(0)$ and $\int_{\mathbb{R}^k} \eta_\epsilon = 1$.

Define the mollification of $\psi_k$ as $h_{k,\epsilon}(r, t) := \int_{\mathbb{R}^k} \eta_\epsilon(y) \psi_k(r - y, t) dy$. Theorem 7 from Appendix C.5 of Evans (2010) asserts that $h_{k,\epsilon} \in C^\infty(\mathbb{R}^k)$. Note that

$$|h_{k,\epsilon}(r, t) - h_{k,\epsilon}(\bar{r}, s)| = \left| \int_{\mathbb{R}^k} \eta_\epsilon(y)(\psi_k(r - y, t) - \psi_k(\bar{r} - y, s)) dy \right|$$

$$\leq \int_{\mathbb{R}^k} \eta_\epsilon(y) |\psi_k(r - y, t) - \psi_k(\bar{r} - y, s)| dy$$

$$\leq \int_{\mathbb{R}^k} \eta_\epsilon(y) C_\varphi \left( \|r - \bar{r}\|_{\mathbb{R}^k}^2 + (t - s)^2 \right)^{\frac{1}{2}} dy$$

$$= C_\varphi \left( \|r - \bar{r}\|_{\mathbb{R}^k}^2 + (t - s)^2 \right)^{\frac{1}{2}},$$

and

$$|h_{k,\epsilon}(r, t) - \psi_k(r, t)| = \left| \int_{\mathbb{R}^k} \eta_\epsilon(y)(\psi_k(r - y, t) - \psi_k(r, t)) dy \right|$$

$$\leq \int_{B_\epsilon^k(0)} \eta_\epsilon(y) C_\varphi \|y\|_{\mathbb{R}^k} dy$$

$$\leq C_\varphi \epsilon.$$

Therefore, $h_{k,\epsilon}$ is also $C_\varphi$-Lipschitz and uniformly converges to $\psi_k$ at a rate $C_\varphi \epsilon$ when $\epsilon \to 0$.

We can then extract a sequence of smooth functions $\psi_{k,n}$ approximating $\psi_k$ by letting $\epsilon_n = \frac{1}{C_\varphi n}$ and $\psi_{k,n} = h_{k,\epsilon_n}$. Note that $\psi_{k,n} \in C^\infty(\mathbb{R}^k)$ is $C_\varphi$-Lipschitz and $\|\psi_{k,n} - \psi_k\|_\infty \leq \frac{1}{n}$.

Let $\chi : \mathbb{R} \to \mathbb{R} \in C^\infty(\mathbb{R})$ be a smooth cutoff function such that $\chi(s) = 1$ on $s \in [-1, 1]$, $\chi(s) \in [0, 1]$ on $s \in (-4, -1) \cup (1, 4)$, and $\chi(s) = 0$ on $(-\infty, -4] \cup [4, \infty)$. Note that $\chi$ and its derivatives of any order are supported on $[-4, 4]$.

We are now ready to define the sequence $\varphi_k : \mathcal{H} \times [0,1] \to \mathbb{R}$ by

$$\varphi_k(x,t) = \chi\left(\frac{\|P_k(x)\|_{\mathbb{R}^k}^2}{k^2}\right)\psi_{k,k}(P_k(x),t).$$

Hence, $\varphi_k(\cdot,t) \in \mathcal{A}$ since $u \to \chi\left(\frac{\|u\|_{\mathbb{R}^k}^2}{k^2}\right)$ and $\psi_{k,k}$ are smooth, $u \to \chi\left(\frac{\|u\|_{\mathbb{R}^k}^2}{k^2}\right)$ is compactly supported, and $\varphi_k$ depends on $x$ only through $P_k(x)$. It remains to verify the following assertions:

1. $\varphi_k$ is uniformly bounded.

2. $\varphi_k$ is uniformly Lipschitz.

3. $\varphi_k$ converges to $\varphi$ pointwisely.

4. $\limsup_k L_{\varphi_k(\cdot,t)}(x) \le L_{\varphi(\cdot,t)}(x), \forall(x,t) \in \mathcal{H} \times [0,1]$.

**1.**

$$|\varphi_k(x,t)| \le |\psi_{k,k}(P_k(x),t)| \le |\psi_k(P_k(x),t)| + \frac{1}{k} \le \|\varphi\|_\infty + 1$$

Given the boundedness of $\varphi$, $\varphi_k(x,t)$ is uniformly bounded.

**2.** Note that

$$\forall u \in \mathbb{R}^d, \nabla_u \chi\left(\frac{\|u\|_{\mathbb{R}^k}^2}{k^2}\right) = \dot\chi\left(\frac{\|u\|_{\mathbb{R}^k}^2}{k^2}\right)\frac{2u}{k^2},$$

where $\dot\chi$ denotes the derivative of $\chi$, and that $\nabla_u\chi\left(\frac{\|u\|_{\mathbb{R}^k}^2}{k^2}\right) = 0$ if $\|u\|_{\mathbb{R}^k}^2 > 4k^2$. When $\|u\|_{\mathbb{R}^k}^2 \le 4k^2$, $\left\|\nabla_u\chi\left(\frac{\|u\|_{\mathbb{R}^k}^2}{k^2}\right)\right\|_{\mathbb{R}^k} \le \|\dot\chi\|_\infty\frac{2}{k^2}\|u\|_{\mathbb{R}^k} \le \|\dot\chi\|_\infty\frac{4}{k}$. Therefore,

$$
\begin{aligned}
|\varphi_k(x,t) - \varphi_k(y,s)| &= \left|\chi\left(\frac{\|P_k(x)\|_{\mathbb{R}^k}^2}{k^2}\right)\psi_{k,k}(P_k(x),t) - \chi\left(\frac{\|P_k(y)\|_{\mathbb{R}^k}^2}{k^2}\right)\psi_{k,k}(P_k(y),s)\right| \\
&\le \left|\chi\left(\frac{\|P_k(x)\|_{\mathbb{R}^k}^2}{k^2}\right)\right||\psi_{k,k}(P_k(x),t) - \psi_{k,k}(P_k(y),s)| \\
&\quad + \left|\chi\left(\frac{\|P_k(x)\|_{\mathbb{R}^k}^2}{k^2}\right) - \chi\left(\frac{\|P_k(y)\|_{\mathbb{R}^k}^2}{k^2}\right)\right||\psi_{k,k}(P_k(y),s)| \\
&\le C_\varphi\left(\|P_k(x) - P_k(y)\|_{\mathbb{R}^k}^2 + |t-s|^2\right)^{\frac{1}{2}} \\
&\quad + \sup_{u \in \mathbb{R}^k}\left\|\nabla_u\chi\left(\frac{\|u\|_{\mathbb{R}^k}^2}{k^2}\right)\right\|_{\mathbb{R}^k}|\psi_{k,k}(P_k(y),s)|\|P_k(x) - P_k(y)\|_{\mathbb{R}^k} \quad (8) \\
&\le C_\varphi\left(\|P_k(x) - P_k(y)\|_{\mathbb{R}^k}^2 + |t-s|^2\right)^{\frac{1}{2}} \\
&\quad + \|\dot\chi\|_\infty\frac{4}{k}\|P_k(x) - P_k(y)\|_{\mathbb{R}^k}(\|\varphi\|_\infty + 1) \quad (9) \\
&\le (C_\varphi + 4\|\dot\chi\|_\infty(\|\varphi\|_\infty + 1))\left(\|P_k(x) - P_k(y)\|_{\mathbb{R}^k}^2 + |t-s|^2\right)^{\frac{1}{2}} \\
&\le (C_\varphi + 4\|\dot\chi\|_\infty(\|\varphi\|_\infty + 1))\left(\|x - y\|_{\mathcal{H}}^2 + |t-s|^2\right)^{\frac{1}{2}},
\end{aligned}
$$

where we apply mean value theorem in $\mathbb{R}^k$ in (8) and the uniform boundedness of $k \to \psi_{k,k}$ in (9).

**3.** Fix $(x,t) \in \mathcal{H} \times [0,1]$. We can find sufficiently large $k$ such that $\|x\|_{\mathcal{H}} < k$. So $\|P_k(x)\|_{\mathbb{R}^k}^2 < k^2$ and $\chi\left(\frac{\|P_k(x)\|_{\mathbb{R}^k}^2}{k^2}\right) = 1$. Hence,

$$
\begin{aligned}
|\varphi_k(x,t) - \varphi(x,t)| &= |\psi_{k,k}(P_k(x),t) - \varphi(x,t)| \\
&\leq |\psi_{k,k}(P_k(x),t) - \psi_k(P_k(x),t)| + |\psi_k(P_k(x),t) - \varphi(x,t)| \\
&\leq \|\psi_{k,k} - \psi_k\|_\infty + |\varphi(\pi_k(x),t) - \varphi(x,t)| \\
&\leq \frac{1}{k} + C_\varphi \|\pi_k(x) - x\|_{\mathcal{H}}.
\end{aligned}
$$

Take $k \to \infty$ and pointwise convergence follows.

**4.** Fix $(x,t) \in \mathcal{H} \times [0,1]$ and $\epsilon \in (0, \frac{1}{2})$. We can find an integer $k_0 > 1$ such that $\|x\|_{\mathcal{H}} < \frac{k_0}{2}$, $\epsilon_{k_0} = \frac{1}{C_\varphi k_0} < \epsilon$.

Then $\forall u \in B_\epsilon^{\mathcal{H}}(x)$, $\|u\|_{\mathcal{H}} \leq \|x\|_{\mathcal{H}} + \epsilon < k_0$, where $B_\epsilon^{\mathcal{H}}(x)$ is the $\epsilon$-ball around $x$ in $\mathcal{H}$. $\forall k > k_0$ and $\forall u \in B_\epsilon^{\mathcal{H}}(x)$, $\chi\left( \frac{\|P_k(u)\|_{\mathbb{R}^k}^2}{k^2} \right) = 1$. Therefore,

$$
\begin{aligned}
L_{\varphi_k(\cdot,t)}(x,\epsilon) &= \sup_{u,v \in B_\epsilon^{\mathcal{H}}(x)} \frac{|\varphi_k(u,t) - \varphi_k(v,t)|}{\|u - v\|_{\mathcal{H}}} \\
&= \sup_{u,v \in B_\epsilon^{\mathcal{H}}(x)} \frac{|\psi_{k,k}(P_k(u),t) - \psi_{k,k}(P_k(v),t)|}{\|u - v\|_{\mathcal{H}}} \\
&= \sup_{u,v \in B_\epsilon^{\mathcal{H}}(x)} \frac{|\psi_{k,k}(P_k(u),t) - \psi_{k,k}(P_k(v),t)|}{\|P_k(u) - P_k(v)\|_{\mathbb{R}^k}} \frac{\|P_k(u) - P_k(v)\|_{\mathbb{R}^k}}{\|u - v\|_{\mathcal{H}}} \\
&\leq \sup_{u,v \in B_\epsilon^{\mathcal{H}}(x)} \frac{|\psi_{k,k}(P_k(u),t) - \psi_{k,k}(P_k(v),t)|}{\|P_k(u) - P_k(v)\|_{\mathbb{R}^k}} \\
&\leq \sup_{u,v \in B_\epsilon^k(P_k(x))} \frac{|\psi_{k,k}(u,t) - \psi_{k,k}(v,t)|}{\|u - v\|_{\mathbb{R}^k}} \quad (10) \\
&= \sup_{u,v \in B_\epsilon^k(P_k(x))} \frac{\left| \int_{B_{\epsilon_k}^k(0)} \eta_{\epsilon_k}(y)(\psi_k(u-y,t) - \psi_k(v-y,t)) dy \right|}{\|u - v\|_{\mathbb{R}^k}} \\
&\leq \int_{B_{\epsilon_k}^k(0)} \eta_{\epsilon_k}(y) \sup_{u,v \in B_\epsilon^k(P_k(x))} \frac{|\psi_k(u-y,t) - \psi_k(v-y,t)|}{\|(u-y) - (v-y)\|_{\mathbb{R}^k}} dy \\
&\leq \sup_{u,v \in B_{\epsilon+\epsilon_k}^k(P_k(x))} \frac{|\psi_k(u,t) - \psi_k(v,t)|}{\|u - v\|_{\mathbb{R}^k}} \quad (11) \\
&\leq \sup_{u,v \in B_{\epsilon+\epsilon_k}^{\mathcal{H}}(\pi_k(x))} \frac{|\varphi(u,t) - \varphi(v,t)|}{\|u - v\|_{\mathcal{H}}} \quad (12) \\
&= L_{\varphi(\cdot,t)}(\pi_k(x), \epsilon + \epsilon_k) \\
&\leq L_{\varphi(\cdot,t)}(\pi_k(x), 2\epsilon), \quad (13)
\end{aligned}
$$

where (10) follows from the fact that $\|P_k(u) - P_k(x)\|_{\mathbb{R}^k} \leq \|u - x\|_{\mathcal{H}} < \epsilon$, and (11) holds because $(u - y) \in B_{\epsilon+\epsilon_k}^k(P_k(x))$. To justify (12), note that $\forall u, v \in B_\epsilon^k(P_k(x))$, we have

$$
\left\| \sum_{i=1}^k u_i e_i - \pi_k(x) \right\|_{\mathcal{H}} = \|u - P_k(x)\|_{\mathbb{R}^k} \leq \epsilon,
$$

$$
\left\| \sum_{i=1}^k u_i e_i - \sum_{i=1}^k v_i e_i \right\|_{\mathcal{H}} = \|u - v\|_{\mathbb{R}^k},
$$

$$
\varphi\left( \sum_{i=1}^k u_i e_i, t \right) = \psi_k(u,t), \varphi\left( \sum_{i=1}^k v_i e_i, t \right) = \psi_k(v,t),
$$

*i.e.*, $\forall u, v \in B_\epsilon^k(P_k(x))$ we can pick two points from $B_\epsilon^{\mathcal{H}}(\pi_k(x))$ such that the fractions in (11) and (12) achieve the same value. Lastly, (13) follows from our choice of $k_0$.

Therefore, $\forall k > k_0$,

$$L_{\varphi_k(\cdot,t)}(x, \epsilon) \le L_{\varphi(\cdot,t)}(\pi_k(x), 2\epsilon).$$

Let $\epsilon \to 0^+$ and we have

$$L_{\varphi_k(\cdot,t)}(x) \le L_{\varphi(\cdot,t)}(\pi_k(x)).$$

Apply upper-semicontinuity of $x \to L_\varphi(\cdot, t)$ and we have

$$\limsup_k L_{\varphi_k(\cdot,t)}(x) \le \limsup_k L_{\varphi(\cdot,t)}(\pi_k(x)) \le L_{\varphi(\cdot,t)}(x).$$

$\square$

### A.3 SUPERPOSITION PRINCIPLES

In this section, we leverage the abstract framework of Theorem 3.4 in Stepanov and Trevisan (2017) to establish the superposition principle in a separable Hilbert space. Intuitively, the principle states that any narrowly continuous curve of probability measures solving the continuity equation can be represented as a distribution over paths that solve the associated ODE. In this way, a statement about marginal laws is transformed into an ODE statement about sample trajectories. The former specifies the object that neural networks are trained to satisfy implicitly, while the latter specifies the mechanism through which new data are sampled with a trained model. By working directly with curves, we avoid introducing a Radon–Nikodym derivative between conditional and marginal laws as required in the FFM framework (Kerrigan et al., 2024). This pathwise view allows us to bypass the absolute-continuity assumption inherent to FFM.

#### A.3.1 FROM CONTINUITY EQUATION TO ODE SOLUTIONS

Let $C([0, 1]; \mathcal{H})$ be the space of continuous function from $[0, 1]$ to $\mathcal{H}$. Let $E_t : z \in C([0, 1]; \mathcal{H}) \to z(t) \in \mathcal{H}$ be the evaluation map, *i.e.*, $E_t(z) = z(t)$. Let $(\cdot)_\#$ denote the pushforward. Specifically, if $z \in C([0, 1]; \mathcal{H})$ has law $\eta$, *i.e*, a probability measure on the space $C([0, 1]; \mathcal{H})$, then $((E_t)_\# \eta)(A) = \eta(z \in C([0, 1]; \mathcal{H}) : E_t(z) = z(t) \in A)$ for a measurable set $A$ in $\mathcal{H}$. Recall that $z : [0, 1] \to \mathcal{H}$ is an absolutely continuous curve if and only if $z$ is differentiable almost everywhere on $[0, 1]$ and $\dot{z}$ is in $L_1([0, 1]; \mathcal{H})$ (Remark 1.1.3 of Ambrosio et al. (2008)). We first define the narrow continuity of measures.

**Definition 15.** *Let $\{\mu_t\}$ be a family of Borel probability measures on a separable Hilbert space $\mathcal{H}$, indexed by $t \in [0, 1]$. We say that $\{\mu_t\}$ is a narrowly continuous curve if and only if the map $t \mapsto \int f \, d\mu_t$ is continuous for all bounded continuous functions $f : \mathcal{H} \to \mathbb{R}$.*

We say a probability measure $\eta$ is concentrated on a set $A$ if $\eta(A) = 1$.

Now we move on to invoke Theorem 3.4 of Stepanov and Trevisan (2017) to develop the superposition principle.

**Theorem 16.** *Let $\{\mu_t\}$ be a narrowly continuous curve of Borel probability measures on the separable Hilbert space $\mathcal{H}$. Let $v_t$ be Borel vector fields such that $\int_{[0,1]} \int_{\mathcal{H}} \|v_t(x)\| d\mu_t dt < \infty$. Suppose $(v_t, \mu_t)$ solves the continuity equation in the sense that for all $\varphi \in \mathcal{A}$,*

$$\frac{d}{dt} \int \varphi \, d\mu_t = \int \langle \nabla\varphi, v_t \rangle d\mu_t. \tag{14}$$

*There exists a probability measure $\eta$ over $C([0, 1]; \mathcal{H})$ concentrated on absolutely continuous curves $z : [0, 1] \to \mathcal{H}$ such that $(E_t)_\# \eta = \mu_t$ for every $t \in [0, 1]$ and one has*

$$\frac{d}{dt} \varphi(z(t)) = (V_t \varphi)(z(t)) \tag{15}$$

*for $\eta$-a.e. $z$ and a.e. $t \in [0, 1]$, for all $\varphi \in \mathcal{A}$.*

*Proof.* Recall that the operator $V_t$ is defined by $(V_t\varphi)(x) = \langle\nabla\varphi(x), v_t(x)\rangle$. This directly satisfies condition (3.1) of Stepanov and Trevisan (2017).

Note that constant functions are members of $\mathcal{A}$.

By Lemma 12, we have $|(V_t\varphi)(x)| = |\langle\nabla\varphi(x), v_t(x)\rangle| \leq \|v_t(x)\| L_\varphi(x)$, which verifies Inequality (3.2) in Stepanov and Trevisan (2017).

The assumption $\int_{[0,1]} \int_{\mathcal{H}} \|v_t(x)\| d\mu_t dt < \infty$ corresponds exactly to condition (3.3) of Stepanov and Trevisan (2017).

Condition (3.4) requires that the continuity equation (14) is satisfied by $\{\mu_t\}$ in the sense of distributions on $[0, 1]$, which holds by assumption.

Finally, Lemma 14 establishes that assumption $\mathcal{A}_1$ in Stepanov and Trevisan (2017) is satisfied.

With all required conditions verified, we may invoke the superposition principle, Theorem 3.4 of Stepanov and Trevisan (2017), so $\eta$ exists as a finite Borel measure. Notice that $1 = \mu_t(\mathcal{H}) = \eta(\{z \in C([0,1]; \mathcal{H}) : z(t) \in \mathcal{H}\}) = \eta(C([0,1]; \mathcal{H}))$. So $\eta$ is indeed a probability measure. $\square$

To apply the superposition principle, we will show that Equation (14) is satisfied by the expected velocity field $v^{\mathbb{X}}$ and the marginal distributions of the process $X_t$. This will allow us to conclude that $\eta$ corresponds to the law of the induced rectified flow. Before proceeding, we reformulate the ODE (15) into a more familiar form in the following corollary.

**Corollary 17.** *Let $\{\mu_t\}$ be a narrowly continuous curve of Borel probability measures on the separable Hilbert space $\mathcal{H}$. Let $v_t$ be Borel vector fields such that $\int_{[0,1]} \int_H \|v_t(x)\| d\mu_t dt < \infty$. Suppose $(v_t, \mu_t)$ solves the continuity equation in a sense that for all $\varphi \in \mathcal{A}$,*

$$\frac{d}{dt}\int \varphi d\mu_t = \int \langle\nabla\varphi, v_t\rangle d\mu_t.$$

*There exists a probability measure $\eta$ over $C([0,1]; \mathcal{H})$ concentrated on absolutely continuous curves $z : [0,1] \to \mathcal{H}$ such that $(E_t)_{\#}\eta = \mu_t$ for every $t \in [0,1]$ and one has*

$$z(t) = z(0) + \int_0^t v_s(z(s))ds$$

*for $\eta$-a.e. $z$ and $\forall t \in [0,1]$.*

*Proof.* Fix $R \in \mathbb{R}_{>0}$. Let $\chi_R \in C^\infty(\mathbb{R})$ such that $\chi_R(s) = 1$ if $|s| \leq R$ and $\chi_R(s) = 0$ if $|s| \geq 2R$. We denote its derivative by $\dot{\chi}_R$. Let $\{e_k\}_{k=1}^\infty$ be an orthonormal basis of $\mathcal{H}$.

Fix $k$. Let $\varphi_{k,R}(x) = \chi_R(\langle x, e_k\rangle)\langle x, e_k\rangle$. Note that $\varphi_{k,R} \in \mathcal{A}$ and if $|\langle x, e_k\rangle| < R$,

$$\nabla\varphi_{k,R}(x) = \chi_R(\langle x, e_k\rangle)e_k + \dot{\chi}_R(\langle x, e_k\rangle)\langle x, e_k\rangle e_k$$
$$= e_k.$$

By applying Theorem 16, for $\eta$-a.e. $z$ and a.e. $t \in [0, 1]$,

$$\frac{d}{dt}\varphi_{k,R}(z(t)) = (V_t\varphi_{k,R})(z(t)).$$

If $|\langle z(t), e_k\rangle| < R$, following the chain rule on Hilbert space (Theorem 2.1 of Coleman (2012)),

$$\frac{d}{dt}\varphi_{k,R}(z(t)) = \langle\dot{z}(t), \nabla\varphi_{k,R}(z(t))\rangle$$
$$= \langle\dot{z}(t), e_k\rangle, \tag{16}$$

and

$$(V_t\varphi_{k,R})(z(t)) = \langle\nabla\varphi_{k,R}(z(t)), v_t(z(t))\rangle$$
$$= \langle e_k, v_t(z(t))\rangle. \tag{17}$$

For each fixed $k$ and $R$, we denote

$$N_{k,R} = \left\{ (z,t) : \dot{z}(t) \text{ doesn't exist or } \frac{d}{dt}\varphi_{k,R}(z(t)) \neq (V_t\varphi_{k,R})(z(t)) \right\}.$$

Note $N_{k,R}$ has measure 0. Let $R_n = n$ and set $N_\infty := \cup_{k=1}^\infty \cup_{n=1}^\infty N_{k,R_n}$. Hence, $N_\infty$, as a countable union of null sets, is also of measure 0 under $\eta \otimes \lambda$, where $\lambda$ is the Lebesgue measure on $[0,1]$. Let $(z,t) \notin N_\infty$. For each $k$, we can choose $n$ such that $R_n = n > |\langle z(t), e_k \rangle|$. By (16) and (17), for $\eta$-a.e. $z$ and a.e. $t \in [0,1]$, $\langle \dot{z}(t), e_k \rangle = \langle e_k, v_t(z(t)) \rangle$ for all positive integers $k$.

So $\dot{z}(t) = v_t(z(t))$ for $\eta$-a.e. $z$ and a.e. $t \in [0,1]$. Therefore, for $\eta$-a.e. $z$, $\forall t \in [0,1]$, $z(t) = z(0) + \int_0^t v_s(z(s))ds$. $\qquad\square$

### A.3.2 VERIFICATION OF ASSUMPTION

Before we move on to the proof of the main theorem, we show that Assumption 2 holds under mild conditions. Recall that $\Phi$ is the solution map defined in Assumption 2. The law of the random process $z_t$ described by $\dot{z}(t) = v_t(z(t))$ with initial value $z(0) \sim \mu_0$ is $\Phi_\#\mu_0$, i.e., for a Borel measurable set $B \subset C([0,1]; \mathcal{H})$, $(\Phi_\#\mu_0)(B) = \mu_0\left(\Phi^{-1}(B)\right)$.

**Proposition 18.** *Let $v_t(\cdot)$ be continuous in an open set $\mathcal{U} \supset [0,1] \times \mathcal{H}$. If there exists a continuous $k(\cdot) \in L^1((0,1))$ such that*

$$\|v_t(x_1) - v_t(x_2)\| \leq k(t)\|x_1 - x_2\|, \forall t \in (0,1), x_1, x_2 \in \mathcal{H},$$

*then Assumption 2 holds.*

*Proof.* The existence and uniqueness are implied by Theorem 16.1 and Theorem 16.2 of Pata (2019).

Let $u_i \in \mathcal{H}$ and $z_i = u_i + \int_0^t v(s, z_i(s))ds$, $i \in 1,2$. Then

$$\|z_1(t) - z_2(t)\|_\mathcal{H} \leq \|u_1 - u_2\|_\mathcal{H} + \int_0^t k(s)\|z_1(s) - z_2(s)\|_\mathcal{H}ds.$$

Applying Gronwall's inequality, $\forall t \in [0,1]$,

$$\|z_1(t) - z_2(t)\|_\mathcal{H} \leq \|u_1 - u_2\|_\mathcal{H} \exp\left(\int_0^t k(s)ds\right).$$

Hence,

$$\sup_{t \in [0,1]} \|z_1(t) - z_2(t)\|_\mathcal{H} \leq \|u_1 - u_2\|_\mathcal{H} \exp\left(\int_0^1 k(s)ds\right).$$

Therefore, the solution map is continuous. $\qquad\square$

While there is no Lipschitz guarantee for generic neural networks, modern architectures can be designed to satisfy global Lipschitz conditions. Examples include spectral normalization Miyato et al. (2018), exact convolution bounds Sedghi et al. (2019), and Jacobian regularization Gouk et al. (2021).

### A.3.3 THE LAW OF ODE SOLUTIONS

We verify that the law $\eta$ produced by the superposition principle in Corollary 17 coincides with the law of the random process described by $\dot{z}(t) = v_t(z(t))$ with initial value $z(0) \sim \mu_0$.

**Theorem 19.** *Let $v_s$ be a Borel vector field on $\mathcal{H}$ and consider the absolutely continuous curves $z : [0,1] \to \mathcal{H}$ satisfying*

$$z(t) = z(0) + \int_0^t v_s(z(s))ds, \forall t \in [0,1]. \tag{18}$$

*Suppose Assumption 2 holds for $(s,z) \to v_s(z)$ with the solution map denoted by $\Phi$ and $\{\mu_t\}$ is a narrowly continuous curve of Borel probability measures on the separable Hilbert space $\mathcal{H}$. If there exists a probability measure $\eta$ over $C([0,1]; \mathcal{H})$ concentrated on the curves $z$ satisfying (18) such that $(E_t)_\#\eta = \mu_t$ for every $t \in [0,1]$, then $\eta = \Phi_\#\mu_0$.*

*Proof.* Let $A \subset C([0, 1]; \mathcal{H})$ be a measurable set.

$$\eta(A) = \eta(A \cap \Phi(\mathcal{H})) \tag{19}$$
$$= \eta(\{\gamma \in \Phi(\mathcal{H}) : \gamma(0) \in \Phi^{-1}(A)\})$$
$$= \eta(\{\gamma \in \Phi(\mathcal{H}) : E_0(\gamma) \in \Phi^{-1}(A)\})$$
$$= ((E_0)_{\#}\eta)(\Phi^{-1}(A)) \tag{20}$$
$$= \mu_0(\Phi^{-1}(A))$$
$$= \Phi_{\#}\mu_0(A)$$

We have (19) and (20) because $\eta$ concentrates on the solutions of the ODE, *i.e.*, $\eta$ is of 0 measure outside $\Phi(\mathcal{H})$ . $\qquad\square$

### A.4 PROOF OF MAIN THEOREM

Now we're ready to prove the main theorem. Given a stochastic process $\mathbb{X}$, we let $v^{\mathbb{X}}$ be the expected velocity defined in Definition 1.

**Theorem 20** (Restatement of Theorem 5)**.** *Assume $\mathbb{X}$ is rectifiable and $\{Z_t\}$ is its induced rectified flow. Then $Z_t = X_t$ in distribution for all $t \in [0, 1]$.*

*Proof.* Recall that $\mathcal{A}$ is the cylindrical functions on $\mathcal{H}$. Let $\mathbb{E}_{\mathbb{X}}$ be the integral *w.r.t.* the path measure of process $\mathbb{X}$. $\mathbb{E}_{X_t}$ is the expectation *w.r.t.* the marginal $X_t$. $\forall \varphi \in \mathcal{A}$,

$$\frac{d}{dt}\mathbb{E}_{\mathbb{X}}[\varphi(X_t)] = \lim_{h \to 0} \frac{\mathbb{E}_{\mathbb{X}}[\varphi(X_{t+h})] - \mathbb{E}_{\mathbb{X}}[\varphi(X_t)]}{h}$$
$$= \lim_{h \to 0} \mathbb{E}_{\mathbb{X}}\left[\frac{\varphi(X_{t+h}) - \varphi(X_t)}{h}\right]$$
$$= \mathbb{E}_{\mathbb{X}}\left[\lim_{h \to 0} \frac{\varphi(X_{t+h}) - \varphi(X_t)}{h}\right] \tag{21}$$
$$= \mathbb{E}_{\mathbb{X}}\left[\frac{d}{dt}\varphi(X_t)\right]$$
$$= \mathbb{E}_{\mathbb{X}}\left[\left\langle \nabla\varphi(X_t), \dot{X}_t \right\rangle\right] \tag{22}$$
$$= \mathbb{E}_{X_t}\mathbb{E}_{\mathbb{X}|X_t}\left[\left\langle \nabla\varphi(X_t), \dot{X}_t \right\rangle\right]$$
$$= \mathbb{E}_{X_t}\left[\left\langle \nabla\varphi(X_t), \mathbb{E}_{\mathbb{X}|X_t}\left[\dot{X}_t\right] \right\rangle\right] \tag{23}$$
$$= \mathbb{E}_{X_t}\left[\left\langle \nabla\varphi(X_t), v^{\mathbb{X}}(X_t, t) \right\rangle\right].$$

Note that following the mean-value theorem (Corollary 3.2 of Coleman (2012)) and Assumption 3,

$$\left|\frac{\varphi(X_{t+h}) - \varphi(X_t)}{h}\right| \le \|\nabla\varphi\|_{\infty} \left\|\frac{X_{t+h} - X_t}{h}\right\| \le \|\nabla\varphi\|_{\infty} \sup_{s \in [0,1]} \left\|\dot{X}_s\right\| < \infty$$

So for all $h$, $\left|\frac{\varphi(X_{t+h}) - \varphi(X_t)}{h}\right|$ is bounded above and (21) follows the dominated convergence theorem (Theorem E.6. of Cohn (2013)). (22) follows the chain rule (Theorem 2.1 of Coleman (2012)). (23) is a direct consequence of Theorem E.11 of Cohn (2013).

Let $\mu_t$ denote the marginal distribution of $X_t$. Then, $\frac{d}{dt}\mathbb{E}_{\mathbb{X}}[\varphi(X_t)] = \int \langle \nabla\varphi(x), v^{\mathbb{X}}(x, t) \rangle d\mu_t(x)$. One the other hand, $\frac{d}{dt}\mathbb{E}_{\mathbb{X}}[\varphi(X_t)] = \frac{d}{dt}\int \varphi(x)d\mu_t(x)$. So $(v^{\mathbb{X}}(x, t), \mu_t)$ solves the continuity equation in Corollary 17. Hence, there exists a probability measure $\eta$ on $C([0, 1]; \mathcal{H})$ concentrated on absolutely continuous curves such that $(E_t)_{\#}\eta = \mu_t$ for every $t \in [0, 1]$ and

$$z(t) = z(0) + \int_0^t v^{\mathbb{X}}(s, z(s))ds.$$

Theorem 19 guarantees that $\eta = \Phi_{\#}\mu_0$, which is the law of the rectified flow $\{Z_t\}$. Therefore, $(E_t)_{\#}\eta = \mu_t$ implies that $Z_t = X_t$ in distribution. $\qquad\square$

## A.5 PROBABILITY FLOW ODE

In this section, we review Gaussian measures in separable Hilbert spaces as a prerequisite for the proof of Proposition 7.

### A.5.1 GAUSSIAN MEASURE IN HILBERT SPACE

A linear operator $Q$ on $\mathcal{H}$ is called *positive* if $\langle Qx, x \rangle \geq 0$ for all $x \in \mathcal{H}$, and *strictly positive* if $\langle Qx, x \rangle > 0$ for all $x \in \mathcal{H} \setminus \{0\}$. Let $\{\zeta_i\}$ be an orthonormal basis of $\mathcal{H}$, and let $Q$ be a positive bounded linear operator. The trace of $Q$ is defined as $\mathrm{Tr}(Q) = \sum_{k=1}^{\infty} \langle Q\zeta_k, \zeta_k \rangle$, which is independent of the choice of orthonormal basis. A self-adjoint, positive, bounded, linear operator $Q$ is called *trace class* if and only if $\mathrm{Tr}(Q) < \infty$.

Following Da Prato and Zabczyk (2014), a Borel probability measure $\mu$ on $\mathcal{H}$ is called *Gaussian* if for every $h \in \mathcal{H}$ and Borel set $B \subset \mathbb{R}$, the pushforward measure $B \mapsto \mu(\{x \in \mathcal{H} : \langle x, h \rangle \in B\})$ is a Gaussian measure on $\mathbb{R}$. Any such Gaussian measure on $\mathcal{H}$ admits a unique mean vector $m \in \mathcal{H}$ and a unique self-adjoint positive bounded linear operator $Q$ such that

$$\mathbb{E}_{x \sim \mu}[\langle h, x \rangle] = \langle m, h \rangle, \quad \text{and} \quad \mathbb{E}_{x \sim \mu}[\langle h_1, x - m \rangle \langle h_2, x - m \rangle] = \langle Qh_1, h_2 \rangle,$$

for all $h, h_1, h_2 \in \mathcal{H}$ (Da Prato and Zabczyk, 2014). The operator $Q$ is called the *covariance operator* of a Gaussian measure on $\mathcal{H}$, and it is always trace class (see Section 2.3.1 of (Da Prato and Zabczyk, 2014)). When $Q$ is a covariance operator, there exists a complete orthonormal system $\{\zeta_k\}_{k=1}^{\infty}$ of eigenvectors of $Q$ with corresponding eigenvalues $\{\lambda_k\}_{k=1}^{\infty}$ (Conway, 1990; Na et al., 2025).

We define the Cameron–Martin space following Pidstrigach et al. (2024). For a Gaussian measure $\mu = \mathcal{N}(0, Q)$ on a Hilbert space $\mathcal{H}$ with a strictly positive $Q$, the Cameron–Martin space $\mathcal{H}_Q$ is defined as

$$\mathcal{H}_Q := Q^{1/2}(\mathcal{H}), \quad \text{with inner product} \quad \langle f, g \rangle_{\mathcal{H}_Q} = \langle Q^{-1/2}f, Q^{-1/2}g \rangle_{\mathcal{H}}.$$

For completeness, we include the definition of an $\mathcal{H}$-valued $Q$-Wiener process:

**Definition 21.** *Let $Q$ be a trace-class, positive, self-adjoint operator on $\mathcal{H}$. A stochastic process $W = \{W_t\}_{t \in [0,T]}$ taking values in $\mathcal{H}$ is called a* standard $Q$-Wiener process *if:*

1. *$W_0 = 0$,*

2. *$W$ has continuous trajectories,*

3. *$W$ has independent increments,*

4. *$W_t - W_s \sim \mathcal{N}(0, (t-s)Q)$ for all $0 \leq s \leq t \leq T$.*

### A.5.2 PROOF OF PROPOSITION 7

**Lemma 22.** *Let $Q$ be a strictly positive trace-class covariance operator and $\kappa \in \mathbb{R}_{>0}$. Let $\mu$ be a Gaussian measure with mean $m$ and a strictly positive trace-class covariance operator $C = \kappa Q$. Then,*

$$\rho_{\mathcal{H}_Q}^{\mu}(x) = \frac{m - x}{\kappa}$$

**Remark 23.** *Note that although $m - x \in \mathcal{H}$ does not necessarily belong to $\mathcal{H}_Q$, $\left\langle \rho_{\mathcal{H}_Q}^{\mu}(x), h \right\rangle_{\mathcal{H}_Q}$ is still well-defined for $h \in \mathcal{H}_Q$. This has been mentioned briefly in Remark 2 of Helin and Burger (2015) and Remark 8 of Lasanen (2012). To see this in more detail, without loss of generality, assume $m = 0$. Let $\{\zeta_i\}$ be an orthonormal basis of $\mathcal{H}$ consisting of eigenvectors of $Q$, with corresponding*

*eigenvalues be $\{\lambda_i\}$. Let $N_1, N_2$ be integers and $N_2 > N_1$. Then,*

$$\mathbb{E}_{z \sim \mu}\left[\left|\left\langle \sum_{i=N_1+1}^{N_2} \frac{\langle z, \zeta_i\rangle_{\mathcal{H}}}{\sqrt{\lambda_i}}\zeta_i, Q^{-\frac{1}{2}}h \right\rangle_{\mathcal{H}}\right|^2\right]$$

$$=\mathbb{E}_{z \sim \mu}\left[\left|\left\langle \sum_{i=N_1+1}^{N_2} \frac{\langle z, \zeta_i\rangle_{\mathcal{H}}}{\sqrt{\lambda_i}}\zeta_i, \sum_{i=1}^{\infty}\left\langle Q^{-\frac{1}{2}}h, \zeta_i\right\rangle_{\mathcal{H}}\zeta_i \right\rangle\right|^2\right]$$

$$=\mathbb{E}_{z \sim \mu}\left[\left|\sum_{i=N_1+1}^{N_2} \frac{\langle z, \zeta_i\rangle_{\mathcal{H}}}{\sqrt{\lambda_i}}\left\langle Q^{-\frac{1}{2}}h, \zeta_i\right\rangle_{\mathcal{H}}\right|^2\right]$$

$$=\sum_{i,j=N_1+1}^{N_2}\mathbb{E}_{z \sim \mu}\left[\frac{\langle z, \zeta_i\rangle_{\mathcal{H}}\langle z, \zeta_j\rangle_{\mathcal{H}}}{\sqrt{\lambda_i\lambda_j}}\left\langle Q^{-\frac{1}{2}}h, \zeta_i\right\rangle_{\mathcal{H}}\left\langle Q^{-\frac{1}{2}}h, \zeta_j\right\rangle_{\mathcal{H}}\right]$$

$$=\sum_{i,j=N_1+1}^{N_2}\left[\frac{\langle C\zeta_i, \zeta_j\rangle_{\mathcal{H}}}{\sqrt{\lambda_i\lambda_j}}\left\langle Q^{-\frac{1}{2}}h, \zeta_i\right\rangle_{\mathcal{H}}\left\langle Q^{-\frac{1}{2}}h, \zeta_j\right\rangle_{\mathcal{H}}\right]$$

$$=\kappa\sum_{i,j=N_1+1}^{N_2}\left[\frac{\lambda_i\langle \zeta_i, \zeta_j\rangle_{\mathcal{H}}}{\sqrt{\lambda_i\lambda_j}}\left\langle Q^{-\frac{1}{2}}h, \zeta_i\right\rangle_{\mathcal{H}}\left\langle Q^{-\frac{1}{2}}h, \zeta_j\right\rangle_{\mathcal{H}}\right]$$

$$=\kappa\sum_{i=N_1+1}^{N_2}\left|\left\langle Q^{-\frac{1}{2}}h, \zeta_i\right\rangle_{\mathcal{H}}\right|^2$$

*Note that $Q^{-\frac{1}{2}}h \in \mathcal{H}$. So $\sum_{i=1}^{n}\left\langle Q^{-\frac{1}{2}}h, \zeta_i\right\rangle_{\mathcal{H}}\zeta_i$ is Cauchy. Hence, $\left\langle \sum_{i=1}^{n}\frac{\langle z,\zeta_i\rangle_{\mathcal{H}}}{\sqrt{\lambda_i}}\zeta_i, Q^{-\frac{1}{2}}h\right\rangle_{\mathcal{H}}$ is Cauchy in $L_2(\mu)$. Hence, for $z \in \mathcal{H}$, $\langle z, h\rangle_{\mathcal{H}_Q} := \lim_n \left\langle \sum_{i=1}^{n}\frac{\langle z,\zeta_i\rangle_{\mathcal{H}}}{\sqrt{\lambda_i}}\zeta_i, Q^{-\frac{1}{2}}h\right\rangle_{\mathcal{H}}$ is well-defined in $L_2(\mu)$.*

*Proof.* It is sufficient to show

$$\int_{\mathcal{H}}\partial_h F(x)d\mu(x) = -\int_{\mathcal{H}}F(x)\langle\kappa^{-1}(m-x), h\rangle_{\mathcal{H}_Q}d\mu(x) \tag{24}$$

Recall that $\{\zeta_i\}$ is an orthonormal basis of $\mathcal{H}$ consisting of eigenvectors of $Q$. Let the corresponding eigenvalues be $\{\lambda_i\}$, and $F$ is a cylindrical function. Define the projection operator $P_k : \mathcal{H}_Q \to \mathbb{R}^d$, $P_k(x) = [\langle x, \zeta_1\rangle, \ldots, \langle x, \zeta_k\rangle]^{tr}$. Let $Y = P_k(X)$ where $X \sim \mathcal{N}(0, C)$. Then $Y \sim \mu^{(k)} := \mathcal{N}(m^{(k)}, \Sigma^{(k)})$ where $m^{(k)} \in \mathbb{R}^k$, $\Sigma^{(k)} \in \mathbb{R}^{k \times k}$, $m_i^{(k)} = \langle m, \zeta_i\rangle$, and $\Sigma_{ij}^{(k)} = \langle C\zeta_j, \zeta_i\rangle$, following Lemma 2.2.2 of Bogachev (1998) and the characterizations of Gaussian measure on Hilbert space in Na et al. (2025). We first evaluate the Gâteaux differential $\partial_h F(x)$:

$$\partial_h F(x) = \lim_{\delta \to 0}\frac{F(x+\delta h) - F(x)}{\delta}$$

$$= \lim_{\delta \to 0}\frac{f(P_k(x+\delta h)) - f(P_k(x))}{\delta}$$

$$= \lim_{\delta \to 0}\frac{f(P_k(x) + \delta P_k(h)) - f(P_k(x))}{\delta}$$

$$= \partial_{P_k(h)}f(P_k(x))$$

$$= \langle\nabla f(P_k(x)), P_k(h)\rangle_{\mathbb{R}^k},$$

where the last equality follows the property of Gâteaux differential (Chapter 2 appendix, (Coleman, 2012)).

Hence,

$$
\begin{aligned}
\int_{\mathcal{H}} \partial_h F(x) d\mu(x) &= \int_{\mathcal{H}} \langle \nabla f(P_k(x)), P_k(h) \rangle_{\mathbb{R}^k} d\mu(x) \\
&= \int_{\mathbb{R}^k} \langle \nabla f(y), P_k(h) \rangle_{\mathbb{R}^k} d\mu^{(k)}(y) \\
&= \langle \mathbb{E}_{Y \sim \mu^{(k)}} [\nabla f(Y)], P_k(h) \rangle_{\mathbb{R}^k}.
\end{aligned}
$$

Recall that $\Sigma_{ij}^{(k)} = \langle C\zeta_j, \zeta_i \rangle = \lambda_i \mathbb{1}_{i=j}$ and $C$ is strictly positive. So $\Sigma^{(k)}$ is invertible. By applying Stein's Identity (Liu and Wang, 2016), we have

$$
\mathbb{E}_{Y \sim \mu^{(k)}} [\nabla f(Y)] = [\Sigma^{(k)}]^{-1} \mathbb{E}_{Y \sim \mu^{(k)}} [f(Y)(Y - m^{(k)})].
$$

Hence,

$$
\begin{aligned}
\int_{\mathcal{H}} \partial_h F(x) d\mu(x) &= \langle \mathbb{E}_{Y \sim \mu^{(k)}} [\nabla f(Y)], P_k(h) \rangle_{\mathbb{R}^k} \\
&= \langle [\Sigma^{(k)}]^{-1} \mathbb{E}_{Y \sim \mu^{(k)}} [f(Y)(Y - m^{(k)})], P_k(h) \rangle_{\mathbb{R}^k} \\
&= \mathbb{E}_{Y \sim \mu^{(k)}} \Big[ f(Y) \langle [\Sigma^{(k)}]^{-1}(Y - m^{(k)}), P_k(h) \rangle_{\mathbb{R}^k} \Big].
\end{aligned}
$$

Now that the left-hand side of (24) has been reduced to an expectation of an inner product in $\mathbb{R}^k$, we apply a similar procedure to the right-hand side. Let $\pi_k h = \sum_{i=1}^{k} \langle h, \zeta_i \rangle_{\mathcal{H}} \zeta_i$ and $h^\perp := h - \pi_k h$. Then,

$$
\begin{aligned}
& \mathbb{E}_{X \sim \mu} \Big[ F(X) \langle \kappa^{-1}(X - m), h \rangle_{\mathcal{H}_Q} \Big] \\
=& \mathbb{E}_{X \sim \mu} \Big[ F(X) \langle \kappa^{-1}(X - m), \pi_k h \rangle_{\mathcal{H}_Q} \Big] + \mathbb{E}_{X \sim \mu} \Big[ F(X) \langle \kappa^{-1}(X - m), h^\perp \rangle_{\mathcal{H}_Q} \Big].
\end{aligned}
$$

It is sufficient to prove (24) by showing

$$
\mathbb{E}_{Y \sim \mu^{(k)}} \Big[ f(Y) \langle [\Sigma^{(k)}]^{-1}(Y - m^{(k)}), P_k(h) \rangle_{\mathbb{R}^k} \Big] = \mathbb{E}_{X \sim \mu} \Big[ F(X) \langle \kappa^{-1}(X - m), \pi_k h \rangle_{\mathcal{H}_Q} \Big] \quad (25)
$$

and

$$
\mathbb{E}_{X \sim \mu} \Big[ F(X) \langle \kappa^{-1}(X - m), h^\perp \rangle_{\mathcal{H}_Q} \Big] = 0. \quad (26)
$$

We first show (25). Let $S^{(k)} = \langle \kappa^{-1}(X - m), \pi_k h \rangle_{\mathcal{H}_Q}$. Hence,

$$
\begin{aligned}
S^{(k)} &= \langle \kappa^{-1} Q^{-\frac{1}{2}}(X - m), Q^{-\frac{1}{2}} \pi_k h \rangle_{\mathcal{H}} \\
&= \langle X - m, \kappa^{-1} Q^{-1} \pi_k h \rangle_{\mathcal{H}}.
\end{aligned}
$$

Note that $Q^{-1} \pi_k h$ is well defined as it is supported on only a finite number of eigenvectors. So $S^{(k)}$ is a Gaussian random variable on $\mathbb{R}$, by definition of Gaussian measure on $\mathcal{H}$.

Following the definition of the covariance operator $Q$ (Lemma 2.15 of Da Prato and Zabczyk (2014)),

$$
\begin{aligned}
\text{Cov}(S^{(k)}, Y_j - m_j^{(k)}) &= \mathbb{E}_{X \sim \mu} [\langle X - m, \kappa^{-1} Q^{-1} \pi_k h \rangle_{\mathcal{H}} \langle X - m, \zeta_j \rangle_{\mathcal{H}}] \\
&= \langle \kappa Q(\kappa^{-1} Q^{-1}) \pi_k h, \zeta_j \rangle_{\mathcal{H}} \\
&= \langle \pi_k h, \zeta_j \rangle_{\mathcal{H}} \\
&= \langle h, \zeta_j \rangle_{\mathcal{H}}.
\end{aligned}
$$

Hence, we can apply the conditional mean estimator for a joint Gaussian distribution (Theorem 3 of Holt and Nguyen (2023)),

$$
\begin{aligned}
\mathbb{E}[S^{(k)} \mid Y] &= \mathbb{E}[S^{(k)}] + [\langle h, \zeta_1 \rangle_{\mathcal{H}}, \ldots, \langle h, \zeta_k \rangle_{\mathcal{H}}][\Sigma^{(k)}]^{-1}(Y - m^{(k)}) \\
&= \langle [\Sigma^{(k)}]^{-1}(Y - m^{(k)}), P_k(h) \rangle_{\mathbb{R}^k}.
\end{aligned}
$$

Note $F(X)$ is measurable by $\sigma(Y)$.

$$
\begin{aligned}
\mathbb{E}_{X \sim \mu}[F(X)S^{(k)}] &= \mathbb{E}_{Y \sim \mu^{(k)}}[\mathbb{E}[F(X)S^{(k)} \mid Y]] \\
&= \mathbb{E}_{Y \sim \mu^{(k)}}[f(Y)\mathbb{E}[S^{(k)} \mid Y]] \\
&= \mathbb{E}_{Y \sim \mu^{(k)}}\left[f(Y)\left\langle [\Sigma^{(k)}]^{-1}(Y - m^{(k)}), P_k(h) \right\rangle_{\mathbb{R}^k}\right].
\end{aligned}
$$

It remains to show (26). Similarly, we define $S^\perp := \left\langle \kappa^{-1}(X - m), h^\perp \right\rangle_{\mathcal{H}_Q} = \left\langle \kappa^{-1}Q^{-\frac{1}{2}}(X - m), Q^{-\frac{1}{2}}h^\perp \right\rangle_{\mathcal{H}}$. Then,

$$
\begin{aligned}
S^\perp &= \left\langle \kappa^{-1}\sum_{i=1}^{\infty}\frac{\langle X - m, \zeta_i\rangle_{\mathcal{H}}}{\sqrt{\lambda_i}}\zeta_i, \sum_{i>k}\frac{\langle h^\perp, \zeta_i\rangle_{\mathcal{H}}}{\sqrt{\lambda_i}}\zeta_i \right\rangle_{\mathcal{H}} \\
&= \kappa^{-1}\sum_{i>k}\frac{1}{\lambda_i}\langle X - m, \zeta_i\rangle_{\mathcal{H}}\langle h^\perp, \zeta_i\rangle_{\mathcal{H}}
\end{aligned}
$$

For $j \leq k$,

$$
\begin{aligned}
\operatorname{Cov}(S^\perp, Y_j - m_j^{(k)}) &= \mathbb{E}_{X \sim \mu}\left[\kappa^{-1}\sum_{i>k}\frac{1}{\lambda_i}\langle X - m, \zeta_i\rangle_{\mathcal{H}}\langle h^\perp, \zeta_i\rangle_{\mathcal{H}}\langle X - m, \zeta_j\rangle_{\mathcal{H}}\right] \\
&= \kappa^{-1}\sum_{i>k}\frac{1}{\lambda_i}\langle h^\perp, \zeta_i\rangle_{\mathcal{H}}\mathbb{E}_{X \sim \mu}\left[\langle X - m, \zeta_i\rangle_{\mathcal{H}}\langle X - m, \zeta_j\rangle_{\mathcal{H}}\right] \\
&= \kappa^{-1}\sum_{i>k}\frac{1}{\lambda_i}\langle h^\perp, \zeta_i\rangle_{\mathcal{H}}\langle \zeta_i, \zeta_j\rangle_{\mathcal{H}} \\
&= 0.
\end{aligned}
$$

Again, by Theorem 3 of Holt and Nguyen (2023),

$$
\mathbb{E}[S^\perp \mid Y] = \mathbb{E}[S^\perp] + [\operatorname{Cov}(S^\perp, Y_1 - m_1^{(k)}), \ldots, \operatorname{Cov}(S^\perp, Y_k - m_k^{(k)})][\Sigma^{(k)}]^{-1}(Y - m^{(k)}) = 0.
$$

So $\mathbb{E}_{X \sim \mu}[F(X)S^\perp] = \mathbb{E}_{Y \sim \mu^{(k)}}[\mathbb{E}[f(Y)S^\perp \mid Y]] = \mathbb{E}_{Y \sim \mu^{(k)}}[f(Y)\mathbb{E}[S^\perp \mid Y]] = 0.$

Therefore,

$$
\int_{\mathcal{H}}\partial_h F(x)d\mu(x) = -\int_{\mathcal{H}}F(x)\left\langle \kappa^{-1}(m - x), h \right\rangle_{\mathcal{H}_Q}d\mu(x).
$$

$\square$

We move on to prove Proposition 7.

*Proof of Proposition 7:* Since $\sigma_t$ is bounded, the assumptions of Theorem 7.2 of (Da Prato and Zabczyk, 2014) are satisfied, and therefore the SDE has a unique *mild* solution, which is a random process $\{Y_t\}$ satisfying

$$
Y_t = Y_0 + \int_0^t -\frac{\sigma_s}{2}Y_s ds + \int_0^t \sqrt{\sigma_s}dW_s.
$$

Let $M_t = \exp\left(\int_0^t \frac{\sigma_s}{2}ds\right)$. Let $\{\zeta_k\}$ be a orthonormal basis of $\mathcal{H}$. By Ito's lemma (Theorem 4.32 from (Da Prato and Zabczyk, 2014)),

$$\langle M_t Y_t, \zeta_k \rangle = \langle M_0 Y_0, \zeta_k \rangle + \int_0^t \langle M_s \zeta_k, \sqrt{\sigma_s} dW_s \rangle + \int_0^t \left\langle \frac{dM_s}{ds} Y_s, \zeta_k \right\rangle + \left\langle M_t \zeta_k, -\frac{\sigma_s}{2} Y_s \right\rangle ds$$

$$= \langle Y_0, \zeta_k \rangle + \int_0^t \langle M_s \zeta_k, \sqrt{\sigma_s} dW_s \rangle$$

$$= \left\langle Y_0 + \int_0^t M_s \sqrt{\sigma_s} dW_s, \zeta_k \right\rangle,$$

where the last equality follows Proposition 4.30 from (Da Prato and Zabczyk, 2014).

Therefore, $Y_t = \exp\left(-\frac{1}{2} \int_0^t \sigma_s ds\right) Y_0 + \int_0^t \exp\left(-\frac{1}{2} \int_s^t \sigma_\tau d\tau\right) \sqrt{\sigma_s} dW_s$.

Following Proposition 4.28 of Da Prato and Zabczyk (2014),

$$Y_t \mid Y_0 \sim \mathcal{N}(\eta(t) Y_0, \kappa(t) Q),$$

where $\eta(t) = \exp\left(-\frac{1}{2} \int_0^t \sigma_s ds\right)$ and $\kappa(t) = \int_0^t \exp\left(-\int_s^t \sigma_\tau d\tau\right) \sigma_s ds$.

Recall that $\mu_{t|Y_0}$ is the conditional distribution of $Y_t|Y_0$. Following Lemma 22, we have

$$\rho_{\mathcal{H}_Q}^{\mu_{t|Y_0}}(y) = \frac{\eta(t) Y_0 - y}{\kappa(t)}.$$

Also,

$$Y_t \overset{d}{=} \eta(t) Y_0 + \sqrt{\kappa(t)} U$$

where $U \sim \mathcal{N}(0, Q)$.

Recall that

$$Y_t' = \eta(t) Y_0' + \sqrt{\kappa(t)} U$$

where $Y_0' \sim P_{\text{data}}$ and $U \sim \mathcal{N}(0, Q)$ are sampled independently.

Na et al. (2025) aim to solve

$$\min_{S:[0,1]\times\mathcal{H}\to\mathcal{H}} \int_0^1 \mathbb{E}_{Y_0 \sim P_{\text{data}}} \mathbb{E}_{Y_t \sim \mu_{t|Y_0}} \left\| S(t, Y_t) - \rho_{\mathcal{H}_Q}^{\mu_{t|Y_0}}(Y_t) \right\|^2 dt,$$

where $S : [0,1] \times \mathcal{H}$ to $\mathcal{H}$ is measurable, and in practice parametrized by a neural network. This is equivalent to

$$\min_{S:[0,1]\times\mathcal{H}\to\mathcal{H}} \int_0^1 \mathbb{E}_{Y_0' \sim P_{\text{data}}} \mathbb{E}_{Y_t' \sim \mu_{t|Y_0'}} \left\| S(t, Y_t') - \rho_{\mathcal{H}_Q}^{\mu_{t|Y_0'}}(Y_t') \right\|^2 dt,$$

because the joint distribution $(Y_0, Y_t) \overset{d}{=} (Y_0', Y_t')$.

This is a minimum mean squared error problem whose solution is the conditional mean

$$S^*(t, y) = \mathbb{E}\left[\rho_{\mathcal{H}_Q}^{\mu_{t|Y_0'}}(Y_t') \mid Y_t' = y\right]$$

$$= \mathbb{E}\left[\frac{\eta(t) Y_0' - Y_t'}{\kappa(t)} \mid Y_t' = y\right]$$

$$= \mathbb{E}\left[-\frac{U}{\sqrt{\kappa(t)}} \mid Y_t' = y\right].$$

Note that $\frac{d}{dt}\sqrt{\kappa(t)} = \frac{\dot{\kappa}(t)}{2\sqrt{\kappa(t)}}$, $\dot{\eta}(t) = -\frac{\sigma_t}{2}\eta(t)$, and

$$\kappa(t) = \eta^2(t) \int_0^t \eta^{-2}(s)\sigma_s ds,$$

$$\dot{\kappa}(t) = 2\eta(t)\dot{\eta}(t) \int_0^t \eta^{-2}(s)\sigma_s ds + \eta^2(t)\big(\eta^{-2}(t)\sigma_t\big)$$

$$= -\sigma_t \eta^2(t) \int_0^t \eta^{-2}(s)\sigma_s ds + \sigma_t$$

$$= \sigma_t(1 - \kappa(t)).$$

Hence,

$$\dot{Y}_t' = \dot{\eta}(t)\, Y_0' + \frac{\dot{\kappa}(t)}{2\sqrt{\kappa(t)}}\, U$$

$$= \left(-\frac{\sigma_t}{2}\,\eta(t)\right) Y_0' \;+\; \frac{\sigma_t[1 - \kappa(t)]}{2\sqrt{\kappa(t)}}\, U$$

$$= -\frac{\sigma_t}{2}\big[\eta(t)\, Y_0' + \sqrt{\kappa(t)}\, U\big] \;+\; \frac{\sigma_t}{2}\,\frac{U}{\sqrt{\kappa(t)}}$$

$$= -\frac{\sigma_t}{2}\, Y_t' \;-\; \frac{\sigma_t}{2}\Big(-\frac{U}{\sqrt{\kappa(t)}}\Big).$$

Therefore,

$$\dot{Y}_t' = -\frac{1}{2}\sigma_t Y_t' - \frac{1}{2}\sigma_t\Big(-\frac{U}{\sqrt{\kappa(t)}}\Big).$$

So equation (6) is equivalent to

$$dY_t = -\frac{\sigma_t}{2}(Y_t + S^*(t, Y_t))dt = \mathbb{E}[\dot{Y}_t' \mid Y_t' = Y_t]dt.$$

$\square$

## B    CONVEX TRANSPORT COSTS AND THE STRAIGHTENING EFFECT

In finite-dimensional spaces, rectified flows are known to reduce transportation costs and exhibit a straightening effect, both desirable properties in generative modeling (Liu et al., 2022). These properties extend naturally to the setting of infinite-dimensional Hilbert spaces and follow from the marginal-preserving property established in Theorem 5. The proofs largely mirror those in the finite-dimensional case and involve routine but lengthy bookkeeping. We begin by defining the $\mathrm{RectFlow}(\cdot)$ and $\mathrm{Rectify}(\cdot)$ operations.

**Definition 24.** *Let $X_0$ and $X_1$ be $\mathcal{H}$-valued random variables. A coupling $(X_0, X_1)$ is called* rectifiable *if its linear interpolation process $\mathbb{X} = \{tX_1 + (1-t)X_0 : t \in [0,1]\}$ is rectifiable. In this case, the $\mathbb{Z} = \{Z_t : t \in [0,1]\}$ in Equation (2) is called the* **rectified flow** *of coupling $(X_0, X_1)$, denoted as $\mathbb{Z} = \mathrm{RectFlow}\,((X_0, X_1))$, and $(Z_0, Z_1)$ is called the* rectified coupling *of $(X_0, X_1)$, denoted as $(Z_0, Z_1) = \mathrm{Rectify}\,((X_0, X_1))$.*

For any convex function $c$, The following theorem establishes that the $\mathrm{Rectify}(\cdot)$ operator does not increase the transport cost $\mathbb{E}[c(Z_1 - Z_0)]$, which corresponds to a special case of Monge's formulation of optimal transport (Section 1.2 of Ambrosio et al. (2024)). It generalizes Theorem 3.5 from Liu et al. (2022) to the Hilbert space setting.

**Theorem 25.** *Let $X_0, X_1$ be $\mathcal{H}$-valued random variables such that the pair $(X_0, X_1)$ is rectifiable, and define $(Z_0, Z_1) = \mathrm{Rectify}(X_0, X_1)$. Then for any convex function $c : \mathcal{H} \to \mathbb{R}$,*

$$\mathbb{E}\left[c(Z_1 - Z_0)\right] \leq \mathbb{E}\left[c(X_1 - X_0)\right].$$

*Proof.*

$$
\begin{aligned}
\mathbb{E}\left[c\left(Z_1 - Z_0\right)\right] &= \mathbb{E}\left[c\left(\int_0^1 v^{\mathbb{X}}(t, Z_t)\,\mathrm{d}t\right)\right] \\
&\leq \mathbb{E}\left[\int_0^1 c\left(v^{\mathbb{X}}(t, Z_t)\right)\mathrm{d}t\right] \qquad (27) \\
&= \mathbb{E}\left[\int_0^1 c\left(v^{\mathbb{X}}(t, X_t)\right)\mathrm{d}t\right] \\
&= \mathbb{E}\left[\int_0^1 c\left(\mathbb{E}\left[(X_1 - X_0) \mid X_t\right]\right)\mathrm{d}t\right] \\
&\leq \mathbb{E}\left[\int_0^1 \mathbb{E}\left[c\left(X_1 - X_0\right) \mid X_t\right]\mathrm{d}t\right] \qquad (28) \\
&= \int_0^1 \mathbb{E}\left[c\left(X_1 - X_0\right)\right]\mathrm{d}t \\
&= \mathbb{E}\left[c\left(X_1 - X_0\right)\right],
\end{aligned}
$$

where we apply Jensen's inequality at (27) and (28). $\qquad\square$

Following Liu et al. (2022), we define a coupling $(X_0, X_1)$ to be *straight* if it is a fixed point of the $\mathrm{Rectify}(\cdot)$ operation. Such couplings are practically useful, as their associated process is linear and can be simulated exactly in a single step. As a direct extension of Theorem 3.6 from Liu et al. (2022) to the Hilbert space setting, the following theorem characterizes this property.

**Theorem 26.** *Let $X_0, X_1$ be $\mathcal{H}$-valued random variables such that the pair $(X_0, X_1)$ is rectifiable. Let $X_t = tX_1 + (1-t)X_0$ and $\mathbb{Z} = \mathrm{RectFlow}\left((X_0, X_1)\right)$. The following statements are equivalent.*

1. *$\exists$ a strictly convex function $c : \mathcal{H} \to \mathbb{R}$, such that $\mathbb{E}\left[c\left(Z_1 - Z_0\right)\right] = \mathbb{E}\left[c\left(X_1 - X_0\right)\right]$.*

2. *$(X_0, X_1)$ is a fixed point of $\mathrm{Rectify}(\cdot)$, that is, $(X_0, X_1) = (Z_0, Z_1)$.*

3. *The rectified flow coincides with the linear interpolation process: $\mathbb{X} = \mathbb{Z}$.*

4. *The paths of the linear interpolation $\mathbb{X}$ do not intersect:*

$$V\left((X_0, X_1)\right) := \int_0^1 \mathbb{E}\left[\left\|X_1 - X_0 - \mathbb{E}\left[X_1 - X_0 \mid X_t\right]\right\|^2\right]\mathrm{d}t = 0.$$

*Note that $V\left((X_0, X_1)\right) = 0$ indicates that $X_1 - X_0 = \mathbb{E}\left[X_1 - X_0 \mid X_t\right]$ almost surely when $t \sim \mathrm{Uniform}\left([0,1]\right)$, meaning that the lines passing through each $X_t$ is unique, and hence no linear interpolation paths intersect.*

*Proof.* Clearly, $3 \implies 2 \implies 1$.

$1 \implies 4$: $\mathbb{E}\left[c\left(Z_1 - Z_0\right)\right] = \mathbb{E}\left[c\left(X_1 - X_0\right)\right] \implies$ the equalities are achieved in inequalities (27) and (28). Since $c$ is strictly convex, $(X_1 - X_0)|X_t$ is a constant almost surely in $\mathbb{X}$ and $t \in [0, 1]$. So $\mathbb{E}[X_1 - X_0|X_t] = \mathbb{E}[X_1 - X_0]$ almost surely in $\mathbb{X}$ and $t \in [0, 1]$ and $V(\mathbb{X}) = 0$.

$4 \implies 3$: $V(\mathbb{X}) = 0 \implies \mathbb{E}[X_1 - X_0|X_t] = \mathbb{E}[X_1 - X_0]$ almost surely in $\mu_t$ and $t \in [0, 1]$. We have $\int_0^s (X_1 - X_0)\,\mathrm{d}t = \int_0^s \mathbb{E}\left[X_1 - X_0 \mid X_t\right]\mathrm{d}t = \int_0^s v^{\mathbb{X}}(t, X_t)\,\mathrm{d}t$ for $s \in (0, 1]$. Hence,

$$X_t = X_0 + \int_0^t (X_1 - X_0)\,\mathrm{d}t = X_0 + \int_0^t v^{\mathbb{X}}(s, X_s)\,\mathrm{d}s.$$

By definition, $Z_t$ also satisfy

$$Z_t = X_0 + \int_0^t v^{\mathbb{X}}(s, X_t) \, \mathrm{d}s.$$

By Assumption 2, for any fixed sample $\omega$, the solution is unique, *i.e.*, $\mathbb{X}(\cdot, \omega) = \mathbb{Z}(\cdot, \omega)$. So $\mathbb{X} = \mathbb{Z}$. $\qquad\square$

We next present a Hilbert space analogue of Theorem 3.7 from Liu et al. (2022), which shows that repeated application of the rectified flow progressively improves the alignment of the coupling, resulting in more linear trajectories and reduced path overlap. Following Liu et al. (2022), the straightness of a continuously differentiable process $\mathbb{Z} = \{Z_t\}$ is measured by by

$$S(\mathbb{Z}) := \int_0^1 \mathbb{E}\left[\left\|(Z_1 - Z_0) - \dot{Z}_t\right\|^2\right] dt.$$

**Theorem 27.** *Let $X_0, X_1$ be $\mathcal{H}$-valued random variables such that the pair $(X_0, X_1)$ is rectifiable. Let $\mathbb{Z}^k$ the $k$-th rectified flow of $(X_0, X_1)$, that is, $\mathbb{Z}^{k+1} = \mathrm{RectFlow}\left((Z_0^k, Z_1^k)\right)$ and $(Z_0^0, Z_1^0) = (X_0, X_1)$. Assume each $(Z_0^k, Z_1^k)$ is rectifiable for $k = 0, \dots, K$. Then*

$$\sum_{k=0}^{K} S\left(\mathbb{Z}^{k+1}\right) + V\left((Z_0^k, Z_1^k)\right) \le \mathbb{E}\left[\|X_1 - X_0\|^2\right].$$

*Hence, if $\mathbb{E}\left[\|X_1 - X_0\|^2\right] < +\infty$, we have $\min_{k \le K} S\left(\mathbb{Z}^k\right) + V\left((Z_0^k, Z_1^k)\right) = O(1/K)$.*

*Proof.*

$$\begin{aligned}
\|X_1 - X_0\|^2 &= \|X_1 - X_0 - \mathbb{E}[X_1 - X_0 \mid X_t]\|^2 \\
&\quad + 2\langle X_1 - X_0 - \mathbb{E}[X_1 - X_0 \mid X_t], \mathbb{E}[X_1 - X_0 \mid X_t]\rangle \\
&\quad + \|\mathbb{E}[X_1 - X_0 \mid X_t]\|^2.
\end{aligned}$$

Then

$$\mathbb{E}\|X_1 - X_0\|^2 = V((X_0, X_1)) + \int_0^1 \mathbb{E}[\|X_1 - X_0 | X_t\|^2] dt. \qquad (29)$$

Note

$$\left\|(Z_1 - Z_0) - \dot{Z}_t\right\|^2 = \|Z_1 - Z_0\|^2 - 2\left\langle Z_1 - Z_0, \dot{Z}_t\right\rangle + \left\|\dot{Z}_t\right\|^2.$$

By moving the integral over $t$ inside the inner product, we get

$$S(\mathbb{Z}) = \int_0^1 \mathbb{E}\left\|\dot{Z}_t\right\|^2 dt - \mathbb{E}\|Z_1 - Z_0\|^2. \qquad (30)$$

By construction, $\dot{Z}_t = \mathbb{E}[X_1 - X_0 | X_t = Z_t]$. Therefore, $\mathbb{E}\left[\left\|\dot{Z}_t\right\|^2\right] = \mathbb{E}[\|X_1 - X_0 \mid X_t\|^2]$. Hence, by Equations (29) and (30),

$$\mathbb{E}\|X_1 - X_0\|^2 - \mathbb{E}\|Z_1 - Z_0\|^2 = S(\mathbb{Z}) + V((X_0, X_1))$$

Applying it to $\mathbb{Z}^k$,

$$\mathbb{E}\left[\left\|Z_1^k - Z_0^k\right\|^2\right] - \mathbb{E}\left[\left\|Z_1^{k+1} - Z_0^{k+1}\right\|^2\right] = S\left(\mathbb{Z}^{k+1}\right) + V\left((Z_0^k, Z_1^k)\right).$$

Summing over $k$ leads to the desired inequality. $\qquad\square$

A practical implication of Theorem 27 is that single-step sampling becomes feasible after several recursive applications of $\mathrm{RectFlow}(\cdot)$.

## C    EXPERIMENTAL DETAILS

This section provides additional results and implementation details that supplement the main text. We begin with further qualitative results on CelebA to highlight the diversity and fidelity of samples generated by our method. We then outline the architectural configurations and training procedures used in our experiments. The code used for our main experiments is available at `https://github.com/Z-Jianxin/Functional-Rectified-Flow`.

### C.1    ADDITIONAL QUALITATIVE RESULTS ON CELEBA

To further illustrate the generative quality of the Functional Rectified Flow (FRF) model, we present additional samples. Figures 3 through 7 show images generated using the FRF model trained on CelebA $64 \times 64$ dataset. These results highlight the model's ability to consistently produce high-quality and diverse samples across multiple runs.

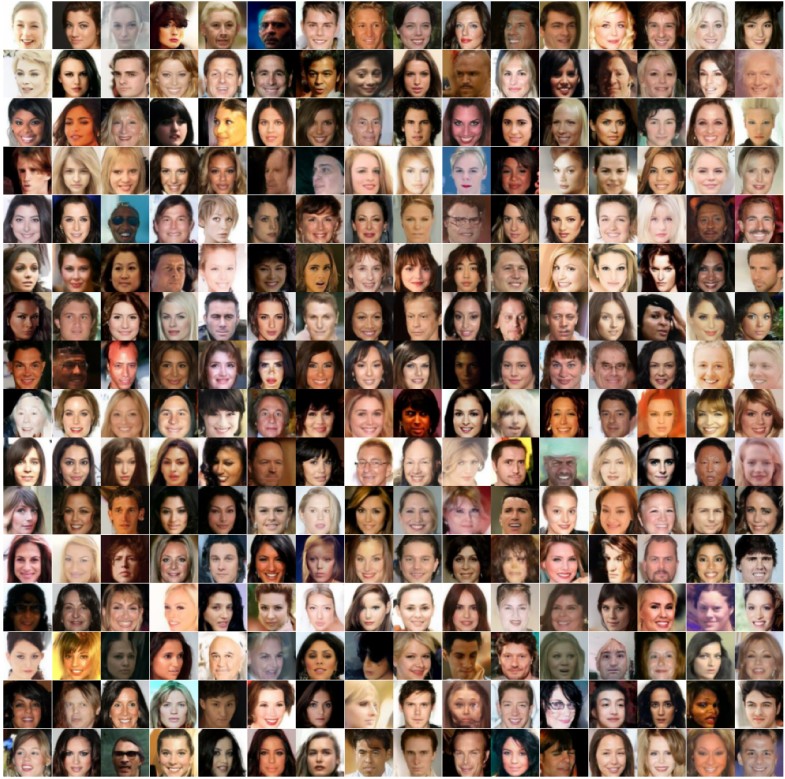

Figure 3: Additional CelebA samples generated by FRF.

### C.2    ARCHITECTURAL DETAILS AND HYPERPARAMETER CHOICES

### C.2.1    INR ON MNIST

We follow Franzese et al. (2023) and adopt the original INR architecture proposed by Sitzmann et al. (2020). The network is a fully connected MLP with 8 layers, each containing 128 neurons and using sinusoidal activations (Sitzmann et al., 2020). We implement the modulation-based meta-learning framework as in Dupont et al. (2022) and Finn et al. (2017). The base network parameters are optimized in the outer loop using the AdaBelief optimizer (Zhuang et al., 2020), with a cosine learning rate schedule ending at $10^{-5}$. The inner loop adapts sample-specific modulation vectors via 3 steps of SGD with a learning rate of $10^{-2}$. Training is performed on 8 NVIDIA A40 GPUs with an effective batch size of $64 \times 8$. The model is trained for $10^6$ steps and takes approximately 3 days.

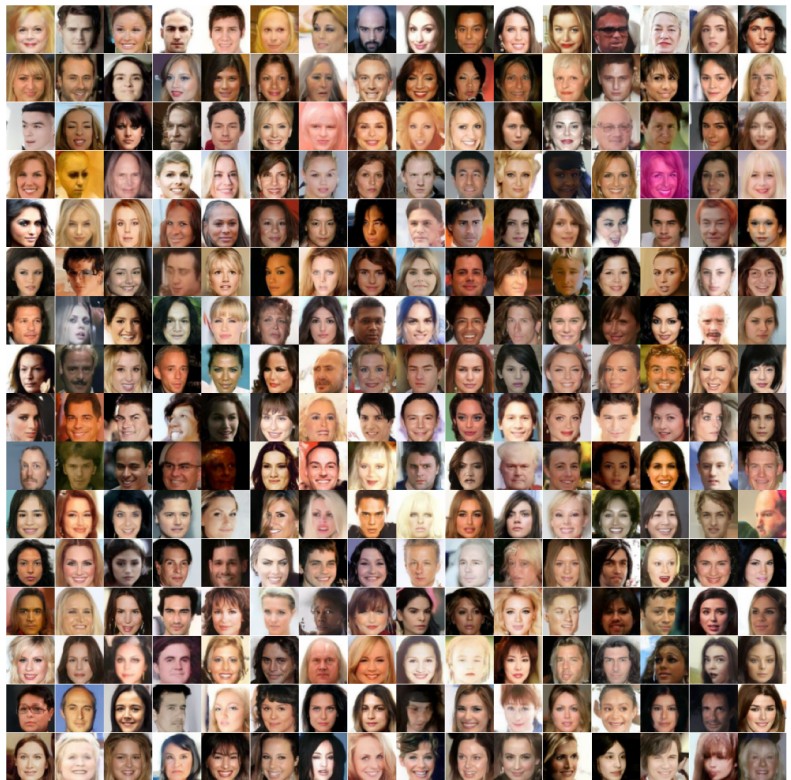

Figure 4: Additional CelebA samples generated by FRF.

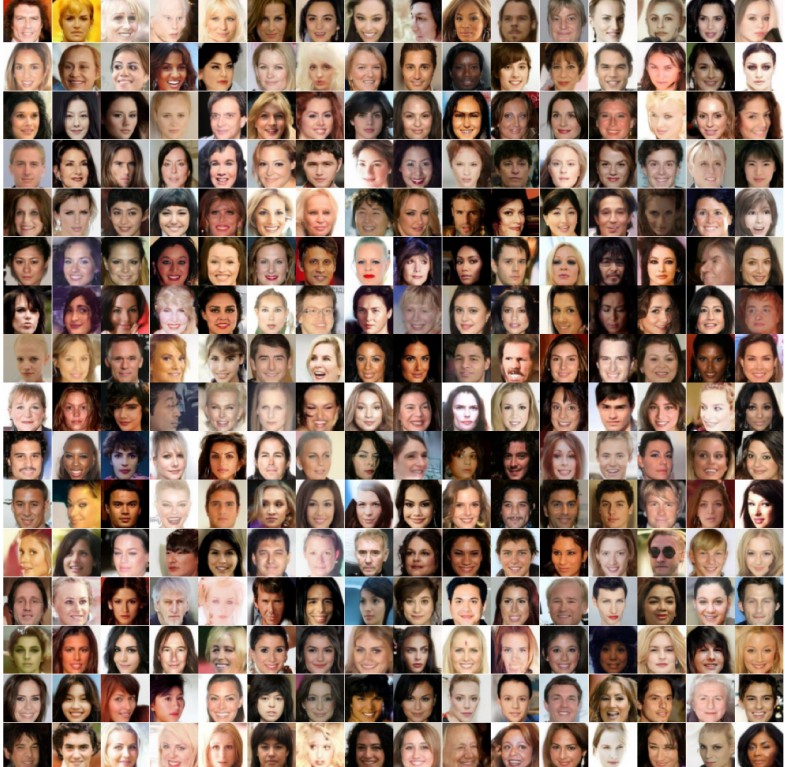

Figure 5: Additional CelebA samples generated by FRF.

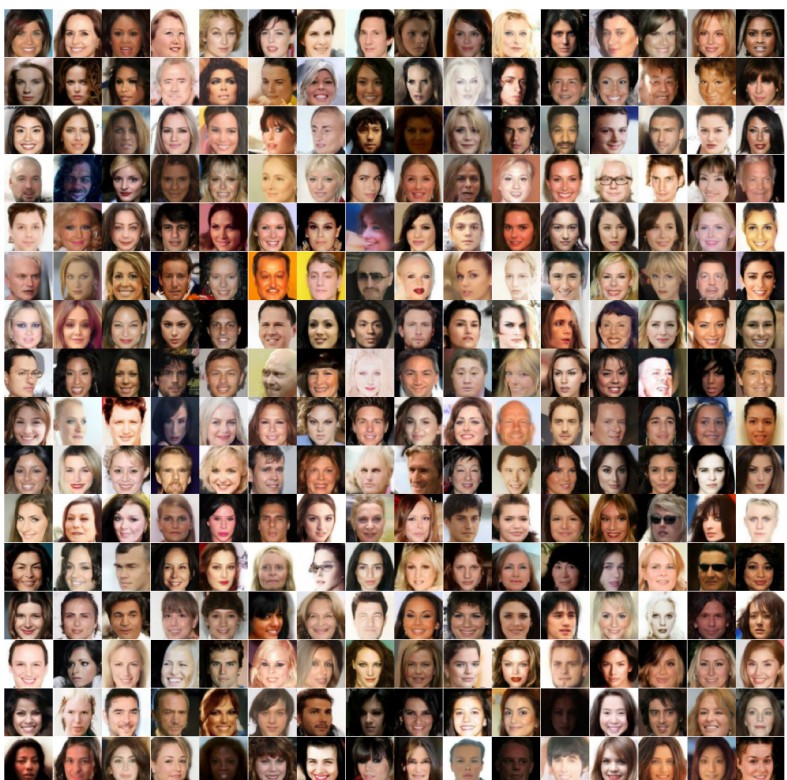

Figure 6: Additional CelebA samples generated by FRF.

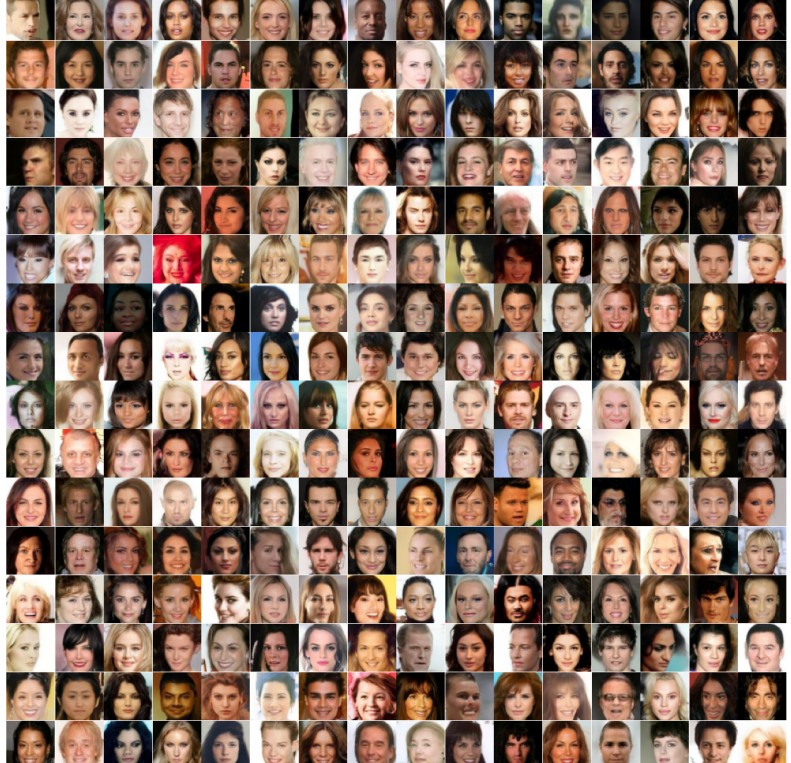

Figure 7: Additional CelebA samples generated by FRF.

### C.2.2 TRANSFORMER ON CELEBA

We follow Franzese et al. (2023) and adopt the UViT backbone introduced in Bao et al. (2023). The model uses 2D sinusoidal positional embeddings, and we set the patch size to 1, effectively treating each pixel as a token. Our architecture matches that of Franzese et al. (2023), consisting of 7 transformer layers, each comprising an 8-head self-attention mechanism and a fully connected feedforward layer. Skip connections are employed following both Franzese et al. (2023) and Bao et al. (2023). For optimization, we use the AdamW optimizer (Loshchilov and Hutter, 2019) with a cosine warm-up schedule, ending at a learning rate of $2 \times 10^{-4}$. The weight decay is set to $10^{-2}$. Training is conducted for $6 \times 10^5$ steps with an effective batch size of 32, and the learning rate is reduced by a factor of 0.1 during the final $10^5$ steps. The model is trained on 8 NVIDIA A40 GPUs, and the full training takes approximately one week.

### C.3 NEURAL OPERATOR ON NAVIER-STOKES DATASET

We follow Kerrigan et al. (2024) and adopt the Fourier Neural Operator (FNO) architecture proposed by Kovachki et al. (2021). The network comprises 4 Fourier layers with 32 modes and 64 hidden channels, along with 256-dimensional lifting and projection layers, mirroring the configuration used in Kerrigan et al. (2024). Optimization is performed using the Adam optimizer with an initial learning rate of $5 \times 10^{-4}$, which is reduced by a factor of 0.1 every 25 epochs. Following Liu et al. (2022), we add a small amount of noise to the input data for smoothing. To mitigate redundancy in the dataset, we follow Kerrigan et al. (2024) and subsample $2 \times 10^4$ datapoints. The model is trained for 300 epochs with a batch size of 128 on a single NVIDIA A100 MIG device, with the full training taking approximately three hours.

## D FAILURE EXAMPLE OF THE FFM ASSUMPTION.

FFM's CM (CM) assumption (Kerrigan et al., 2024) already fails in the fundamental case where $X_0 = X_1 \sim \mathcal{N}(0, C)$ with covariance operator $Ce_n = \frac{1}{n^2}e_n$ for an orthonormal basis $\{e_n\}$ of $L_2$. The CM space of $C$ is $\mathcal{H}_C = \mathrm{Im}(C^{1/2}) = \{x : \sum_{n=1}^{\infty} \langle x, e_n \rangle^2 / \lambda_n < \infty\}$, where $\lambda_n = 1/n^2$ are the eigenvalues of $C$. Thus a sample $x$ lies in the CM space iff

$$\sum_{n=1}^{\infty} \langle x, e_n \rangle^2 n^2 < \infty.$$

For $X \sim \mathcal{N}(0, C)$, the Karhunen–Loève expansion gives $\langle X, e_n \rangle \sim \mathcal{N}(0, \lambda_n)$, so we can write $\langle X, e_n \rangle = \frac{1}{n}\xi_n$ with $\xi_n \sim \mathcal{N}(0, 1)$ i.i.d. Plugging this into the CM condition yields

$$\sum_{n=1}^{\infty} \langle X, e_n \rangle^2 n^2 = \sum_{n=1}^{\infty} \xi_n^2.$$

Since $\xi_n^2$ are i.i.d. with $\mathbb{E}[\xi_n^2] = 1$, the strong law of large numbers implies $\sum_{n=1}^{\infty} \xi_n^2 = \infty$ almost surely, so $\mathbb{P}(X \in \mathcal{H}_C) = 0$. Consequently, even in this trivial setting where no learning is required, a draw $X_1$ from $\mathcal{N}(0, C)$ lies in the CM space with probability zero, and the CM assumption imposed by FFM fails in this case.

## E EMPIRICAL VERIFICATION OF MARGINAL PRESERVATION THEOREM

To provide an intuitive illustration of Theorem 5, we run a synthetic experiment in the Hilbert space $L^2([0, 1])$. We construct a time-dependent process $X_t$ that linearly interpolates between two different random functions: $X_0$, drawn from a smooth Gaussian process with a Matérn-type covariance, and $X_1$, drawn from a mixture of two Gaussian processes with opposite low-frequency mean functions. Using the closed-form expression of the expected velocity associated with this interpolation, we simulate the rectified flow $Z_t$ starting from the same distribution at $t = 0$. Figure 8 shows, for $t \in \{0, 0.25, 0.5, 0.75, 1\}$, shoulder-to-shoulder plots of 512 sample paths from $X_t$ and $Z_t$ as functions on $[0, 1]$. At each time, the cloud of curves generated by the rectified flow is visually indistinguishable from that of the original process, including the emergence and disappearance of the bimodal structure.

## F    DISCLOSURE OF LLM USAGE

Large Language Models (LLMs) were used solely to assist with polishing the writing and presentation of this paper. All technical contributions were developed entirely by the authors without LLM involvement.

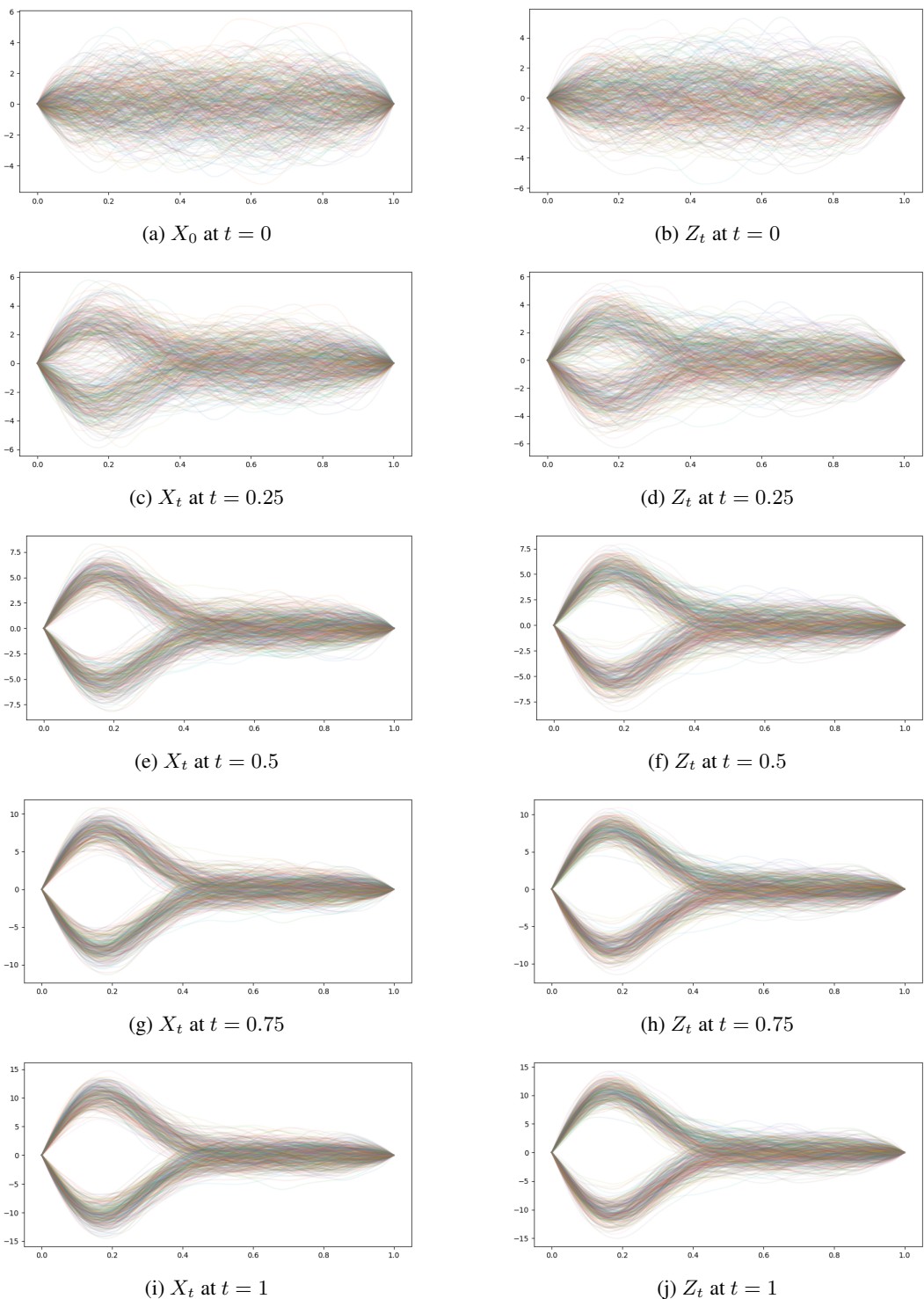

Figure 8: Shoulder-to-shoulder comparison of the original interpolation process $X_t$ (left column) and the rectified flow $Z_t$ (right column) at different times $t \in \{0, 0.25, 0.5, 0.75, 1\}$. $X_0$ is a Gaussian process with Matérn-type covariance and $X_1$ is a mixture of two Gaussian processes. Each panel displays 512 sample paths in $L^2([0, 1])$. The close visual agreement between the left and right panels at each time illustrates the marginal-preserving property $Z_t \overset{d}{\approx} X_t$.

