# OpenReview forum: "Flow Straight and Fast in Hilbert Space: Functional Rectified Flow"
_ICLR.cc/2026/Conference — ICLR 2026 Poster_

### Official Review · Reviewer_CfoQ · 2025-10-24

**Soundness:** 3
**Presentation:** 1
**Contribution:** 2
**Rating:** 4
**Confidence:** 3

**Summary:**

This paper establishes the rectified flow in Hilbert space, by providing a theoretical result proving the superposition principle in separable Hilbert Space. This justifies the extension of the usual flow matching loss in the Hilbert space,  providing an alternative approach to Kerrigan's Functional Flow matching that not only requires the existsnce of the Radon Nykodim derivative but also the target data to live in the Cameron Martin space of the initial Gaussian Measure.
The paper implements the Flow matching in the Hilbert space by embedding the hilbert space element through functional mapping via implicit neural representation / transformers.

**Strengths:**

The paper has a solid contribution regarding the superposition principle in separable Hilbert Space, whose significance may even extend beyond the flow matching.  It provides an alternative to the flow matching techniques like Functional Flow matching in the infinite dimensional space.   The paper also provides valuable insights that the proposed theoretical framework can realize the paths/dynamics proposed on other works (Functional Probability Flow ODE, Functional Flow Matching)

**Weaknesses:**

While the theoretical contributions deserves much credit, but the paper suffers from several weaknesses in its presentations.

### 1.
The theoretical novelty seems orthogonal to the methodological advantage in the current presentation of the paper.  This paper guarantees that "the flow matching of usual type (Lipman et al, Liu et al)" is theoretically justified on the Hilbert space.  However, as a method, (4) on its own is not novel, and it is impossible in the current set of results to conclude whether the performance advantage is coming from theoretical soundness of the proposed approach or from the power of the Hilbert-space embedding machinations like Transformer / INR.
In order to channel the theoretical novelty to practical significance, I believe an extensive, corner case experiment is required to show that FRM overcomes the problem of FFM; for instance, one may come up with the case in which there is no well defined RN derivative, and show that FRM outperforms FFM that is applied as an an approximate procedure ( with the same representation power allowed to both methods).  Some effort is needed to fundamentally isolate the power of the transformer from the threotical novelty.


### 2.
The presentation of the novelty of this work was very hard to follow-- the EFM's theoretical requirements are noted, but to what end? I do agree that these requirements are more impractical than the ones required by the proposed approach, but they have not been shown to emerge as the real problem in practice.  So many methods in machine learning are applied to the real world problem with hand-wavy assumptions --- these ambiguities would be of interest in machine learning only if they present a real threat.     Without the presenting that a "real threat" is triggered by impractical functional assumptions, the motivation of overcoming EFM's assumptions is weak.

At the same time, if the goal of this work is purely theoretical,  I feel that EFM deserves different treatment in this paper, as a more restrictive framework purported to the same theoretical goal, for example.

**Questions:**

Please see the weaknesses section. Does the theoretical soundness of this framework, as a method, provide an advantage over other comparative methods in axis independent from the architectural representation power or way of embedding an element of hilbert space element?

In a similar note, is there any "practical" cases in which X1 lies outside of the Cameron Martin Space. Can you come up with a synthetic case for example to show that the proposed approach is **empirically** superior to functional flow matching? (I guess adding jumps / disconitniuty might do a trick when we set H = L2. )

**Details Of Ethics Concerns:**

Nothing in particular.

---

> ### Author Response · Authors · 2025-11-24
>
> Thanks for the careful review and insights.
>
> W1 & Q1 . isolate the theoretical novelty:
>
>     This is precisely the setup we employed in our experiments. As detailed in Section 6.1.1, both FRF and FDP are equipped with INR using the exact same architecture. In Section 6.1.2, both methods use the same vision transformer. In Section 6.2, all methods utilize the same neural operator. Since the network architectures are identical to the competing methods (adopted from their original papers to avoid bias), the observed performance gain is attributable to the proposed flow matching methodology rather than architectural differences.
>
> W2 & Q2. real threat of FFM's assumption:
>
> FFM's assumption fails even in the simplest case where $X_0 = X_1 \sim \mathcal{N}(0, C)$, with $C$ being the diagonal operator defined by $C e_n = \frac{1}{n^2}e_n$ for an orthonormal basis {$e_n$} in $L_2$ (so $C$ has eigenvalues $\lambda_n = \frac{1}{n^2}$). For a sample $x$ to reside in the Cameron-Martin (CM) space of $C$ (i.e., the image of $C^{1/2}$), the condition is $\sum_{n=1}^{\infty} \frac{\langle x, e_n\rangle^2}{\lambda_n} < \infty$. Substituting $\lambda_n = n^{-2}$, this requires $\sum_{n=1}^{\infty} \langle x, e_n\rangle^2 n^2 < \infty$.
>
> By the Karhunen-Loève expansion [1], the coefficient $\langle x, e_n \rangle$ is a centered Gaussian with variance $\lambda_n = \frac{1}{n^2}$. We can write $\langle x, e_n \rangle = \frac{1}{n}\xi_n$ where $\xi_n \sim \mathcal{N}(0,1)$. Thus, the condition becomes $\sum_{n=1}^{\infty} \xi_n^2 < \infty$. This is a sum of independent Chi-squared variables with mean 1, which diverges to infinity with probability 1. This implies that even in this trivial setting where no learning is required, $X_1$ fails to reside inside the CM space of $C$ with probability 1. Therefore, the assumptions of FFM are mathematically restrictive to the point where they exclude the support of this trivial setting.
>
> Again, we appreciate the reviewers for the insights. We believe the questions raised by the reviewer will improve the clarity of our paper significantly. We will include this discussion in the next revision to highlight our contributions.
>
> [1] https://en.wikipedia.org/wiki/Kosambi%E2%80%93Karhunen%E2%80%93Lo%C3%A8ve_theorem

---

### Official Review · Reviewer_Yrn2 · 2025-10-30

**Soundness:** 3
**Presentation:** 2
**Contribution:** 2
**Rating:** 6
**Confidence:** 3

**Summary:**

The authors demonstrate theoretically the soundness of most general Rectified Flows on infinite-dimensional spaces, greatly relaxing the assumptions imposed by previous works. They propose three ways to train the proposed method, FRF, which they evaluate on two different modalities, on which they significantly improve on the SOTA.

**Strengths:**

- The work is well-anchored in the literature regarding the problem it tackles (flows on infinite-dimensional spaces). Good comparisons are offered at the end of section 4.
- The framework proposed generalises the previous results of the field, and essentially match what we would expect Rectified Flows to be like in this setting; that is to say, they recover essentially the same method, finally, in theory.
- There are good theoretical discussions in the appendix – namely, about how reasonable the assumptions are in this work. In general, the work in the appendix seems very thorough and introduces the topic well.
- The proof of the main theorem seems sound, and is well-organised, in general.

**Weaknesses:**

- On the theoretical part of the paper, there is, I believe, a great problem in the general presentation: as I first read the paper (without the appendix first), I completely missed out on the part about the superposition, and, when I came back to read the abstract, I was surprised to find it there. In general, I would argue that too much of the technical content is hidden in the appendix. Understandably, the content is highly technical and arguably too dense for the main part of a Machine Learning paper. However, essentially omitting main results (included in the abstract itself) seems wrong, and, if space is an issue, other parts (for instance, background) can be trimmed on. I think this is important. Probably some informal/intuitive explanations can be given.
- While the generality of this method on the interpolation choice and the weaknesses of the other methods are pointed out, it required going through the appendix quite thoroughly to get an idea of what the assumptions made in this work were.
- While the work is well-grounded with respect to the generative functional literature, it is less so with respect to the generative modelling/flow-based modelling literature, arguably.
- The empirical validation, I would argue, is the main weakness of this work, and perhaps the method approach in general. It seems that the training objective is pretty standard, so then section 5 must be the novelty, empirically, where different architectures are proposed. Then, in section 6, only one architecture is tried out on each of the datasets. I can imagine that the neural operators can only work in the Navier-Stokes case, but that which worked on MNIST surely can be run on CelebA, unless I am mistaken?
- Such low FID difference on MNIST, especially at such low FIDs too, may not be used as evidence for improvement. I am not even certain it is a good metric for MNIST. (I agree, though, that it shows that the method works.)
- I would refrain from using $\mathcal{O}$ to express “order of” with constants following.

**Questions:**

1. To be certain: what are the main assumptions made in this work?
2. Although they are certainly all essentially equivalent, could you justify formulating this as Rectified Flows (which are not really considered state-of-the-art), instead of, say, Flow Matching [1] or Stochastic Interpolants [2]?
3. How does this work relate to stochastic processes? Is it the case, similar to finite-dimensional (typically Euclidean) data, that we can recover the same marginals without the Wiener process (probability flow ODE [3])?
4. Not being an expert on functional ML, how do you train your models on *images*? And why is it a relevant experiment whatsoever? Is there an advantage to this representation over the typical tensorial CxHxW one?
5. Why did you not try out all applicable methods on the same datasets?

Overall, I think this is a good work, but some clarifications must be made, and mostly the work would benefit a lot from a slightly different presentation, in my opinion.


### References
[1] Yaron Lipman, Ricky T. Q. Chen, Heli Ben-Hamu, Maximilian Nickel, Matt Le. “Flow Matching for Generative Modeling”

[2] Michael S. Albergo, Nicholas M. Boffi, Eric Vanden-Eijnden. “Stochastic Interpolants: A Unifying Framework for Flows and Diffusions”

[3] Yang Song, Jascha Sohl-Dickstein, Diederik P. Kingma, Abhishek Kumar, Stefano Ermon, Ben Poole. “Score-Based Generative Modeling through Stochastic Differential Equations”

---

> ### Author Response · Authors · 2025-11-24
>
> We thank the reviewer for their insightful assessment and for recognizing the theoretical soundness and generality of our framework. We value the constructive feedback regarding the presentation of the superposition principle and the clarity of our assumptions. We address your questions and concerns point-by-point below.
>
> W1. Superposition:
>
>     We appreciate you pointing this out. We agree that the superposition principle is central to the paper's intuition. In the final revision, we will move the discussion and definition of the superposition principle from the appendix to the main paper to ensure these concepts are immediately accessible.
>
> W2. Placement of assumptions:
>
>     We clarify that the framework relies primarily on Assumptions 2 and 3. Assumption 2 requires that the ODE driven by $v$ admits a continuous, unique solution. In Proposition 18, we demonstrate that Assumption 2 holds under standard Lipschitz conditions. We presented Proposition 18 as a supporting result rather than a core assumption because Assumption 2 can technically hold under broader conditions. We will explicitly clarify this distinction in the revised manuscript.
>
> W3. Literature review:
>
>     We will expand the literature review in Section 2 to cover finite-dimensional generative models more comprehensively.
>
> W4. Empirical validation:
>
>     We acknowledge this suggestion and will endeavor to include additional experiments in the final revision, if computational resources permit.
>
>     We clarify that our primary contribution is the theoretical framework rather than novel architectures. We deliberately applied existing architectures (e.g., FNO, ViT) to the specific datasets used in their original proposing papers. This ensures that the performance gains we report are attributable to our method (FRF) rather than architectural modifications.
>
> W5. Margins on MNIST:
>
>     We understand the concern regarding low FID differences. However, reproducing the full suite of image experiments at the scale of large baselines is computationally prohibitive for us. We focused on providing rigorous error bars for the Navier-Stokes experiments (scientific data) where functional methods are most relevant.
>
> W6. Big-O notation:
>
>     We agree with this stylistic point and will remove the informal use of Big-O notation regarding parameter counts in the revision.

---

> > ### Author Response · Authors · 2025-11-24
> >
> > Q1. Main assumptions:
> >
> >     Please refer to our response in W2.
> >
> > Q2. FRF vs Flow Matching [1] or Stochastic Interpolants [2]:
> >
> >     We chose the Rectified Flow formulation for specific theoretical reasons. The ratio of densities used in Flow Matching (Eq. 8 in [1]) translates to the Radon-Nikodym derivative in Functional Flow Matching, which requires strict absolute continuity assumptions to justify its existence. Generalizing Stochastic Interpolants to functional space also has technical hurdles; their proofs rely on PDEs of densities, which require careful translation to Hilbert spaces (e.g., via Gaussian reference measures, as Lebesgue-like measures do not exist in infinite dimensions, or via a purely measure-theoretic equivalence). Our framework avoids these hurdles by posing much more realistic assumptions and utilizing a more straightforward concept, namely, the superposition principle.
> >
> > Q3. Connections to stochastic processes:
> >
> >     Your observation is correct. FRF is designed to recover the same marginals without invoking a Wiener process, analogous to the Probability Flow ODE (PF-ODE) in [3]. In Proposition 7, we show that the PF-ODE is essentially a rectified flow with non-linear interpolants. Moreover, by carefully choosing $\sigma_t$ via Proposition 7, one can recover an SDE whose PF-ODE is the rectified flow.
> >
> > Q4. Images as functions:
> >
> >     In this context, images are modeled as functions mapping coordinates to pixel values (e.g., $I: \mathbb{R}^2 \to \mathbb{R}^3$ for color). One advantage of this parameterization, when equipped with INR, is resolution independence, allowing for effortless super-resolution. The functional perspective also allows mixed precision training/sampling. This leads to a performance gain when equipped with a diffusion-autoencoder as demonstrated in [4].
> >
> > Q5. All applicable methods on the same datasets:
> >
> >     Our experimental design was constrained by computational resources. For instance, training and sampling on MNIST takes approximately 3 days on 8xA40 GPUs, and CelebA takes 7 days. The cost of running every method on every dataset is prohibitive. Our strategy was to align our settings with prior works as closely as possible to maximize the breadth of valid comparisons within our budget. Additionally, transferring architectures across modalities is non-trivial (e.g., Neural Operators require significant modification, like adding convolutional components, to work effectively on image data).
> >
> > We hope these responses address your concerns. We are committed to incorporating these clarifications into the final version of the paper to improve its clarity and completeness.
> >
> >
> > [1] Yaron Lipman, Ricky T. Q. Chen, Heli Ben-Hamu, Maximilian Nickel, Matt Le. “Flow Matching for Generative Modeling”
> >
> > [2] Michael S. Albergo, Nicholas M. Boffi, Eric Vanden-Eijnden. “Stochastic Interpolants: A Unifying Framework for Flows and Diffusions”
> >
> > [3] Yang Song, Jascha Sohl-Dickstein, Diederik P. Kingma, Abhishek Kumar, Stefano Ermon, Ben Poole. “Score-Based Generative Modeling through Stochastic Differential Equations”
> >
> > [4] Sam Bond-Taylor, Chris G. Willcocks, "$\infty$-Diff: Infinite Resolution Diffusion with Subsampled Mollified States"

---

### Official Review · Reviewer_JJQ6 · 2025-10-31

**Soundness:** 2
**Presentation:** 1
**Contribution:** 2
**Rating:** 4
**Confidence:** 4

**Summary:**

This paper extends the **Rectified Flow (RF)** framework to **infinite-dimensional Hilbert spaces**, providing a theoretical foundation for functional generative modeling.
The proposed **Functional Rectified Flow (FRF)** generalizes the original rectified flow (Liu et al., 2022) beyond Euclidean domains, offering a deterministic and mathematically rigorous formulation for *function-valued random variables*.
Overall, the work aims to lift rectified flow theory into infinite-dimensional settings, positioning it as a unifying framework for functional generative modeling.

**Strengths:**

1. The formulation is mathematically solid, carefully extending continuity equations and rectified flows to Hilbert spaces via the *superposition principle*.
2. The work unifies **rectified flow**, **functional flow matching**, and **functional probability flow ODEs** under a single theoretical umbrella, clarifying their relationships.
3. Results show modest but consistent improvement in *FID* and *density MSE* metrics compared to functional diffusion processes (FDP) and functional flow matching (FFM).

**Weaknesses:**

1. The main theoretical advancement—lifting rectified flow to separable Hilbert spaces—is **mathematically correct but conceptually incremental** relative to prior work on functional flow matching (Kerrigan et al., 2024) and functional diffusion models (Franzese et al., 2023). The novelty largely lies in formalizing the *marginal-preserving property* in infinite dimensions rather than introducing fundamentally new generative insights.
2. Experimental improvements (e.g., FID 0.43 → 0.41 on MNIST) are marginal and within statistical noise.  Experiments are confined to relatively small datasets and fixed architectures; no ablation or analysis of functional-space regularization is included.

**Questions:**

1. Can the authors provide more intuition on the *practical implications* of Hilbert-space rectification — e.g., does it enable higher-resolution function generation or better memory scaling?
2. Could the marginal-preserving theorem be empirically verified (e.g., via sample-wise trajectory interpolation analysis)?
3. Is there a computational or convergence benefit of FRF over functional flow matching in terms of training stability or sampling complexity?
4. Would FRF remain valid if the underlying Hilbert space is replaced by a Banach space, or if data lies on mixed discrete–continuous domains?

---

> ### Author Response · Authors · 2025-11-24
>
> We thank the reviewer for the constructive feedback and for recognizing the mathematical solidity of our formulation. We appreciate the opportunity to clarify our contributions and experimental results.
>
>
> W1. Contribution compared to prior works:
>
>     We emphasize that Functional Rectified Flow (FRF) is developed independently of Functional Flow Matching (FFM) and Functional Diffusion Processes (FDP), differing significantly in both derivation and assumptions. Justifying the marginal-preserving property in Hilbert space is also highly non-trivial. We believe the unification of functional rectified flow, functional flow matching, and functional probability flow ODEs under a single theoretical umbrella is a significant generative insight, as mentioned in "strengths".
>
> W2. Experiments:
>
>     We believe the datasets like MNIST and CelebA are not small in academia standards. Producing an error bar for MNIST and CelebA would be beyond our computational capacity. (MNIST takes 3 days on 8 X A40s, and CelebA takes a week on 8 x A40). We prioritized establishing statistical significance on the challenging Navier-Stokes dataset (Section 6.2), where we provide error bars confirming that our improvements are statistically significant.
>
>     As for ablation, in Section 6.1.1, both FRF and FDP are equipped with INR using the exact same architecture. In Section 6.1.2, both methods use the same vision transformer. In Section 6.2, all methods utilize the same neural operator. Since the network architectures are identical to the competing methods (adopted from their original papers to avoid bias), the observed performance gain is attributable to the proposed flow matching methodology rather than architectural differences.
>
>     Could the reviewer clarify what is meant by "functional-space regularization"? We are happy to address this if given more specific context.
>
> Q1. Practical implications of Hilbert-space rectification:
>
>     The Hilbert-space rectification offers two immediate practical benefits. As demonstrated in Section 6.1.1, equipping FRF with models like INRs enables effortless zero-shot super-resolution. The model learns the function, not the pixels, allowing evaluation at higher resolutions. The framework is naturally suited for scientific data (e.g., Navier-Stokes), which are solutions to differential equations and mathematically reside in functional spaces.
>
> Q2. empirical verification of marginal preservation:
>
>     We agree that empirical verification adds value. We can verify Theorem 5 using synthetic distributions (e.g., $X_0, X_1$ sampled from Gaussians with distinct means/covariances) where the ground truth is analytically tractable. We will include this verification in the revision.
>
> Q3.  benefit over FFM:
>
>     While the sampling complexity is comparable to FFM, we observe that FRF generally exhibits greater stability during training.
>
> Q4. Extension to Banach:
>
>     Extending this framework to Banach spaces is an exciting direction for future work. However, it presents significant theoretical challenges; specifically, our current proof of the marginal preservation theorem relies on the basis structure of Hilbert spaces (Parseval's identity, inner products). A Banach space extension would require novel proving techniques. We will add a discussion on these limitations and future directions in the conclusion.
>
> We once again thank the reviewer for their thoughtful comments. We believe these questions have helped us identify areas where the manuscript can be strengthened.

---

### Official Review · Reviewer_yJzb · 2025-11-01

**Soundness:** 3
**Presentation:** 3
**Contribution:** 3
**Rating:** 6
**Confidence:** 3

**Summary:**

This paper presents a rigorous theoretical extension of rectified flows to infinite-dimensional Hilbert spaces, establishing a unified framework for functional generative modeling. The authors prove marginal-preserving property for rectified flows in this settings. It is demonstrated that functional mapping of flows and ODE of probabilistic flows arise as special cases within their framework. Empirically, the method achieves state-of-the-art results on images and scientific datasets using three different neural architectures.

**Strengths:**

- The Hilbert space extension of rectified flows is novel and rigorous, addressing key limitations of prior functional models by relaxing restrictive measure-theoretic assumptions.
- The work successfully unifies several functional generative models, providing a cohesive theoretical perspective.
- Extensive experiments across multiple architectures (INR, Transformer, Neural Operator) and data types demonstrate consistent superiority over competitive baselines.

**Weaknesses:**

- The theory relies on the assumption that the stochastic process $\mathbb X$ is "pathwise continuously differentiable." Although this is more plausible than absolute continuity of measures, it is still a non-trivial assumption about the smoothness of data trajectories. It would be useful to briefly discuss the practical validity of this assumption for real functional data (e.g., images as functions, solutions to partial differential equations). Such a statement is not obvious if, for example, the image contains contrasting edges.
- While the experiments are convincing, they are limited to three datasets. Including a more diverse set of functional data, such as irregularly sampled time series or 3D shapes, would further strengthen the claim of general applicability. Furthermore, a direct comparison of training/sampling speed versus other functional models (FDP, FFM) would be highly informative, given that one of the key motivations for rectified flows is efficiency. Of the three datasets listed, two are datasets with images for which the need to apply this functional technique is not at all obvious -- models built on conventional finite-dimensional principles work well with them.
- The excellent performance is demonstrated by plugging into established, powerful architectures (INR, Transformer, FNO). An open question is how much of the performance gain is due to the proposed theoretical framework versus the inherent capacity of these architectures. An ablation studying a simpler backbone could help isolate the contribution of the rectified flow objective itself.

**Questions:**

- The assumption of pathwise differentiability is central to your theory. How is it implemented or tested during training, especially when using discrete data to approximate functions?
- The paper highlights computational efficiency as a motivation. Could you provide quantitative results (e.g., wall-clock time, number of function evaluations) comparing the sampling speed of your FRF against functional diffusion or flow matching models?
- In sentence 18, you note that Lipschitz's global conditions can be applied to neural networks. Did you use such methods (e.g., spectral normalization) in your experiments to ensure the correctness of the ODE formulation? If not, did you observe any instabilities during numerical integration?

---

> ### Author Response · Authors · 2025-11-24
>
> We thank the reviewer for their constructive feedback and for recognizing the rigor of our theoretical extension to Hilbert spaces, as well as the strength of our empirical results across various architectures. We appreciate the thoughtful questions regarding the differentiability assumptions, which we address below.
>
> W1. Pathwise continuously differentiable assumption:
>
>     We clarify that the pathwise continuous differentiability assumption in Definition 1 applies to the design of the flow, rather than the spatial smoothness of the data distribution itself. Precisely, it requires the trajectory $t \to X_t$ to be differentiable with probability $1$.For example, with a flow design of $X_t = \alpha(t) X_1 + \beta(t)X_0$, the assumption holds as long as the functions $\alpha(t)$ and $\beta(t)$ are continuously differentiable. The specific form used in Rectified Flows, $X_t=tX_1 + (1-t)X_0$, is always continuously differentiable with respect to time $t$.
>
> W2. Dataset diversity:
>
>     We appreciate the suggestion for broader modalities. However, we note that our current evaluation suite is already more diverse than standard baselines in this field, allowing for fair benchmarking. For instance, FDP experiments were limited to MNIST/CIFAR-10, and FFM focused primarily on Navier-Stokes and small tabular datasets. Extending to domains like 3D shapes requires developing novel domain-specific functional architectures (analogous to ViT for images or FNO for PDEs), which we view as a promising direction for future work rather than a prerequisite for validating the current theoretical framework.
>
> W3. Ablation and Architectural Bias:
>
>     Regarding the source of performance gains: In Section 6.1.1 (INR), Section 6.1.2 (ViT), and Section 6.2 (Neural Operator), we strictly utilized the exact same network architectures as the competing methods (FDP and FFM). We deliberately adopted these backbones from the original papers to avoid architectural bias. Consequently, the observed performance improvements are directly attributable to the proposed functional flow matching objective rather than the capacity of the underlying neural networks.
>
> Q1.
>
> See W1.
>
> Q2. Computational efficiency comparisons:
>
> We acknowledge the importance of this comparison. We will endeavor to include quantitative metrics in the final revision if our experimental bandwidth allows.
>
> Q3.Global Lipschitz on neural nets:
>
> We clarify that we did not apply specific techniques (such as spectral normalization) to force Lipschitz conditions, as we aimed to avoid modifying established architectures to ensure fair comparison. The commentary around line 1172 was intended to highlight that neural networks can be designed to satisfy these assumptions if theoretically necessary. Empirically, we did not observe any numerical instabilities during integration in our experiments.
>
> We hope these responses address your concerns. We are committed to incorporating these clarifications into the final version of the paper to improve its clarity and completeness.

---

### Author Response · Authors · 2025-12-04
**Change list in the revision**

**Summary of Revisions and Reviewer Concerns Addressed**

We thank the reviewers (YRN2, YJZB, JJQ6, CFOQ) for their constructive and detailed feedback and the hard work of AC. In response, we have made the following focused improvements:

* **Added a superposition discussion** to the main paper (addresses *YRN2*).
* **Clarified that the pathwise (C1) assumption is a modeling/design choice rather than a property required of the data** (addresses *YJZB*).
* **Expanded the future-work section to mention evaluation on more diverse datasets** (addresses *YJZB*).
* **Strengthened ablation clarity by stating explicitly that all methods are compared under identical architectures**, adopted from the original papers (addresses *YJZB, JJQ6, YRN2, CFOQ*).
* **Revised the contributions to highlight the core generative insight** that our framework *unifies several existing functional generative models* (addresses *JJQ6*).
* **Added empirical verification of marginal preservation** using small synthetic experiments (addresses *JJQ6*).
* **Added an explicit future-work remark on extensions to Banach spaces** (addresses *JJQ6*).
* **Clarified technical assumptions**, including comments on conditions ensuring well-posed IVPs (addresses *YRN2*).
* **Expanded the literature review** with additional citations on standard generative modeling approaches (addresses *YRN2*).
* **Removed unnecessary big-O notation** to streamline exposition (addresses *YRN2*).
* **Added further explanation on modeling images as functions** in the functional generative setting (addresses *YRN2*).
* **Added a discussion of the practical implications and potential risks of the FFM assumption** (addresses *CFOQ*).

---

### Meta-Review · Area_Chair_fKsQ · 2026-01-08

**Summary:**

This paper establishes a rigorous mathematical framework for Rectified Flows in infinite-dimensional Hilbert spaces. By leveraging the superposition principle for continuity equations, it unifies existing functional generative models (e.g., Functional Flow Matching and Functional Probability Flow ODEs) under one umbrella while relaxing restrictive measure-theoretic assumptions.

The initial reviews were borderline and mixed, with two "Marginally Above" (6) and two "Marginally Below" (4) ratings (one of them being flagged as LLM generated). The consensus praised the mathematical rigor and the elegance of the superposition-based derivation. Major concerns focused on the practical necessity of the theoretical contributions (i.e., do the restrictive assumptions of prior work actually matter in practice), the clarity of presentation (key concepts buried in the appendix), and whether empirical gains were due to the method or superior architectures (e.g., INRs, Transformers). There were also requests for more extensive experiments on large datasets.

**Reviewer Concerns:**

**Addressed concerns**:
- Architectural Bias (CfoQ, yJzb, JJQ6): Reviewers questioned if performance gains came from using powerful backbones (INRs, FNOs) rather than the method itself. The authors confirmed they used identical architectures to the baselines (FDP, FFM) , isolating the gains to the flow formulation.
- Theoretical Necessity (CfoQ): Reviewer CfoQ asked if the "Cameron-Martin space" assumption in FFM was a "real threat." The authors provided a concrete mathematical proof that even a trivial Gaussian process fails this assumption, proving the necessity of their relaxation.
- Pathwise Differentiability (yJzb): The reviewer worried that this assumed data smoothness. The authors clarified that this is a property of the flow interpolation (which is designed by the user, e.g., linear interpolation), not the data distribution.

**Partially addressed**:
- Presentation (Yrn2): The authors committed to several action items on improving presentation, such as moving the critical "Superposition Principle" discussion from the appendix to the main text, but some are not addressed fully yet.

**Outstanding concern**:
- Experimental Scope (yJzb, JJQ6, Yrn2): Reviewers requested broader modalities (3D shapes) or larger image datasets. The authors argued their benchmarks match the current "Functional Generative Modeling" literature standards and cited compute constraints. While reasonable, the empirical scope remains limited compared to standard generative model papers.

**Reviewer Scores:**

- Reviewer yJzb (Original: 6): Predicted: 6. The clarification that smoothness assumptions apply to the design, not the data, removes their main theoretical concern, but the lack of broader experiments (which they requested) and run time analysis limits enthusiasm.
- Reviewer JJQ6 (Original: 4): Predicted: 4. This reviewer is generally questioning practical benefit of the theoretical framework, and noted weak empirical improvements (e.g., FID 0.43 vs 0.41 on MNIST). While the authors elaborate on the non-trivialness of their theoretical contribution, there isn't direct new experimental results to address the questions on practical benefits so the reviewer may remain score of 4. Given the LLM Usage flag, I'd down-weight this review.
- Reviewer Yrn2 (Original: 6): Predicted: 6. The presentation fixes promised in the rebuttal are actionable, but wouldn't suffice for further score increase.
- Reviewer CfoQ (Original: 4): Predicted: 6. The rebuttal provided a counter-example showing why prior assumptions fail even in trivial cases. This directly answers their question about "real threat".

---

### Decision · Program_Chairs · 2026-01-26

Accept (Poster)